# Scale Equivariant Graph Metanetworks

**Ioannis Kalogeropoulos**[1,2*]  **Giorgos Bouritsas**[1,2*]  **Yannis Panagakis**[1,2]

[1]National and Kapodistrian University of Athens    [2]Archimedes/Athena RC, Greece

## Abstract

This paper pertains to an emerging machine learning paradigm: learning higher-order functions, i.e. functions whose inputs are functions themselves, *particularly when these inputs are Neural Networks (NNs)*. With the growing interest in architectures that process NNs, a recurring design principle has permeated the field: adhering to the permutation symmetries arising from the connectionist structure of NNs. *However, are these the sole symmetries present in NN parameterizations?* Zooming into most practical activation functions (e.g. sine, ReLU, tanh) answers this question negatively and gives rise to intriguing new symmetries, which we collectively refer to as *scaling symmetries*, that is, non-zero scalar multiplications and divisions of weights and biases. In this work, we propose *Scale Equivariant Graph MetaNetworks - ScaleGMNs*, a framework that adapts the Graph Metanetwork (message-passing) paradigm by incorporating scaling symmetries and thus rendering neuron and edge representations equivariant to valid scalings. We introduce novel building blocks, of independent technical interest, that allow for equivariance or invariance with respect to individual scalar multipliers or their product and use them in all components of ScaleGMN. Furthermore, we prove that, under certain expressivity conditions, ScaleGMN can simulate the forward and backward pass of any input feedforward neural network. Experimental results demonstrate that our method advances the state-of-the-art performance for several datasets and activation functions, highlighting the power of scaling symmetries as an inductive bias for NN processing. The source code is publicly available at https://github.com/jkalogero/scalegmn.

## 1 Introduction

Neural networks are becoming the workhorse of problem-solving across various domains. To solve a task, they acquire rich information which is stored in their learnable parameters throughout training. Nonetheless, this information is often opaque and difficult to interpret. This begs the question: *How can we efficiently process and extract insights from the information stored in the parameters of trained neural networks, and how to do so in a data-driven manner?* In other words, can we devise architectures that learn to process other neural architectures?

The need to address this question arises in diverse scenarios: *NN post-processing*, e.g. analysis/interpretation (i.e. inferring NN properties, such as generalisation and robustness [74, 19]), as well as editing (e.g. model pruning [22], merging [75] or adaptation to new data) - or *NN synthesis* (e.g. for optimisation [12] or more generally parameter prediction/generation [31, 67, 32, 58]). Furthermore, with the advent of *Implicit Neural Representations*,[1] [13, 52, 57, 70, 53] trained NN parameters are increasingly used to represent datapoint *signals*, such as images or 3D shapes, replacing raw representations, i.e., pixel grids or point clouds [18]. Consequently, many tasks involving such data,

---

[*]Equal contribution. Corrrespondence to ioakalogero@di.uoa.gr, g.bouritsas@athenarc.gr

[1]INR: *A coordinate-processing NN "overfitted" on a given signal, e.g. image, 3D shape, physical quantity.*

across various domains such as computer vision [47] and physics [68], which are currently tackled using domain-specific architectures (e.g. CNNs for grids, PointNets [63] or GNNs [60] for point clouds and meshes), could potentially be solved by *NNs that process the parameters of other NNs*.

The idea of processing/predicting NN parameters per se is not new to the deep learning community, e.g. it has been investigated in *hypernetworks* [27] or even earlier [65]. However recent advancements, driven by the need for INR processing, leverage a crucial observation: *NNs have symmetries*. In particular, several works have identified that the function represented by an NN remains intact when applying certain transformations to its parameters [28, 11] with the most well-known transformations being *permutations*. That is, hidden neurons can be arbitrarily permuted within the same layer, along with their incoming and outgoing weights. Therefore, all the tasks mentioned in the previous paragraph can be considered part of *equivariant machine learning* [10], as they involve developing models that are invariant or equivariant to the aforementioned transformations of neural networks. Recently Navon et al. [54] and Zhou et al. [86] acknowledged the importance of permutation symmetries and first proposed equivariant architectures for Feedforward NN (FFNN) processing, demonstrating significant performance improvements against non-equivariant baselines. These works, as well as their improvements [85] and extensions to process more intricate and varying architectures [33, 44, 88], are collectively known as *weight space networks* or *metanetworks*, with *graph metanetworks* being a notable subcase (graph neural networks with message-passing).

Nevertheless, it is known that permutations are *not* the only applicable symmetries. In particular, theoretical results in various works [11, 61], mainly unified in [25] show that FFNNs exhibit additional symmetries, which we collectively here refer to as *scaling symmetries*: multiplying the incoming weights and the bias of a hidden neuron with a non-zero scalar $a$ (with certain properties) while dividing its outgoing weights with another scalar $b$ (often $b = a$), preserves the NN function. Intuitively, permutation symmetries arise from the *graph/connectionist structure* of the NN whereas different scaling symmetries originate from its *activation functions* $\sigma$, i.e. when it holds that $\sigma(ax) = b\sigma(x)$. However, equivariance w.r.t. scaling symmetries has not received much attention so far (perhaps with the exception of *sign symmetries* - $a \in \{-1, 1\}$ [41, 40]), and it remains effectively unexplored in the context of metanetworks.

To address this gap, we introduce in this paper a graph metanetwork framework, dubbed as *Scale Equivariant Graph MetaNetworks- ScaleGMN*, which guarantees equivariance to permutations *and* desired scaling symmetries, and can process FFNNs of arbitrary graph structure with a variety of activation functions. At the heart of our method lie novel building blocks with scale invariance/equivariance properties w.r.t. arbitrary families of scaling parameters. We prove that ScaleGMN can simulate the forward and backward pass of an FFNN for arbitrary inputs, enabling it to reconstruct the function represented by any input FFNN and its gradients.

Our contributions can be summarised as follows:

- We extend the scope of metanetwork design from permutation to scaling symmetries.
- We design invariant/equivariant networks to scalar multiplication of individual multipliers or combinations thereof, originating from arbitrary scaling groups.
- We propose scale equivariant message passing, using the above as building blocks, unifying permutation and scale equivariant processing of FFNNs. Additionally, the expressive power of our method is analysed w.r.t. its ability to simulate input FFNNs.
- Our method is evaluated on 3 activations: ReLU *(positive scale)*, tanh *(sign)* and sine *(sign - we first characterise this using a technique from [25])* on several datasets and three tasks (INR classification/editing & generalisation prediction), demonstrating superior performance against common metanetwork baselines.

## 2 Related work

**Neural network symmetries.** Long preceding the advent of metanetworks, numerous studies delved into the inherent symmetries of neural networks. Hecht-Nielsen [28] studied FFNNs and discovered the existence of permutation symmetries. Chen et al. [11] showed that, in FFNNs with tanh activations, the only function-preserving smooth transformations are permutations and sign flips (multiplication of incoming and outgoing weights with a sign value) - this claim was strengthened in [21]; this was the first identification of scaling symmetries. Follow-up works extended these observations to

other activations, such as *sigmoid and RBF* [37] and *ReLU* [56, 51, 61, 64, 26], and architectures, such as RNNs [3, 2], characterising other scaling symmetries and providing conditions under which permutations and scalings are the only available function-preserving symmetries or the parameters are identifiable given the input-output mapping. Recently, Godfrey et al. [25] provided a technique to characterise such symmetries for arbitrary activations that respect certain conditions, unifying many previous results. Additionally, symmetries have been found in other layers, e.g. batch norm [6, 14, 16] (scale invariance) or softmax [36] (translation invariance). Prior to metanetworks, symmetries were mainly studied to obtain a deeper understanding of NNs or in the context of optimisation/learning dynamics [56, 71, 4, 17, 36, 5, 83, 84] and/or model merging/ensembling [20, 1, 69, 59, 55].

**Metanetworks.** The first solutions proposed for NN processing and learning did not account for NN symmetries. Unterthiner et al. [74] and Eilertsen et al. [19], initially employed standard NNs on vectorised (flattened) CNN weights or some statistics thereof, to predict properties of trained neural networks such as generalization or estimation of a hyperparameter resp.). Similar methods have been proposed to process continuous data represented as INRs. In [78] high-order spatial derivatives are used (suitable only for INRs), in [47] the architecture operates on stacked parameter vectors (but is constrained by the assumption that all INRs are trained from the same initialization), while in [18] and [7] the authors learn low-dimensional INR embeddings (which are further used for downstream tasks) by jointly fitting them with meta-learning. Finally, Schürholt et al. [66] learn representations of NN weights with self-supervised learning and in [67] this is extended to generative models.

In contrast to the above methods, our work follows a recent stream of research focusing on *equivariant metanetworks*. Navon et al. [54] and Zhou et al. [85] first characterised *all linear* equivariant layers to permutation symmetries of MLPs and combined them with non-linearities, while in the latter this was extended to CNNs. These approaches derive intricate weight-sharing patterns but are non-local[2] and cannot process varying architectures. In follow-up works, Zhou et al. [86] constructed equivariant attention layers and in [87] the above characterisation was generalised to arbitrary input architectures and layers (e.g. RNNs and transformers) introducing an algorithm for automatic linear layer (with weight-sharing) construction. A different route was taken in the very recent works of Kofinas et al. [33] and Lim et al. [44], where input NNs are treated as a special class of graphs and are naturally processed with Graph Neural Networks (GNNs), with appropriate symmetry breaking wherever necessary. This perspective has been adopted several times in the deep learning literature [9, 23, 45, 77, 80] and in a few examples, GNNs were applied on the graph representation for, e.g. neural architecture search ([82, 72, 31]), albeit without mapping parameters to the graph edges. In parallel to our work, [73] extended the framework of [85] to incorporate scaling and sign-flipping symmetries. They construct non-local equivariant layers, resulting in an architecture with fewer trainable parameters. Nevertheless, their method suffers from limited expressive power, while their experimental results showcase limited advancements. In contrast to our method, this approach limits the choice of activation functions to those equivariant to the studied symmetries. Finally, their framework cannot be extended to other activation functions, as the parameter sharing must be re-designed from scratch, while it is not suitable for processing diverse architectures.

**Scale Equivariance** Equivariance to vector scaling (or more generally to matrix multiplication with diagonal matrices) remains to date underexplored in the machine learning community. Sign symmetries received attention in the work of Lim et al. [42], where an *invariant* network (*SignNet*) was designed, mainly to process eigenvectors, with the theoretical analysis revealing universality under certain conditions. This was extended in [39], where the importance of sign *equivariance* on several tasks was demonstrated and a sign equivariant network was proposed. In our work, we draw inspiration from these architectures and extend their formulations to arbitrary scaling symmetries.

## 3 Notation and Preliminaries

**Notation.** We denote vectors, matrices, and tensors with bold-face letters, e.g., $\mathbf{x}, \mathbf{X}, \mathbf{X}$, respectively and sets with calligraphic letters $\mathcal{X}$. A normal font notation will be used for miscellaneous purposes (mostly indices and functions). Datapoint (input) functions/signals will be denoted with $u$, while higher-order functions (functions of functions) will be denoted with fraktur font $\mathfrak{F}$.

**Functions of Neural Networks.** Consider functions of the form $u_{G,\boldsymbol{\theta}} : \mathcal{X} \to \hat{\mathcal{X}}$. Each function is parameterised (1) by a **computational graph** $G$, which determines all the mathematical operations

---

[2]They can be perceived as the metanetwork analogue of Invariant Graph Nets [50] for conventional graphs.

that should be performed to evaluate the function $u$ at a datapoint $\mathbf{x} \in \mathcal{X}$. When $u$ is a neural network, $G$ is determined by the NN *architecture*. Additionally, $u$ is parameterised by (2), by **a tuple of numerical parameters $\boldsymbol{\theta}$**, on which the aforementioned mathematical operations are applied (along with the input $\mathbf{x}$) - these are the *learnable parameters*, which can be arranged into a vector. We are interested in learning unknown higher-order functions $\mathfrak{F} : \hat{\mathcal{X}}^{\mathcal{X}} \to \mathcal{Y}$ of the form $\mathfrak{F}\left(u_{G,\boldsymbol{\theta}}\right)$. In our case, the goal is to *learn $\mathfrak{F}$ by accessing solely the parameters $(G, \boldsymbol{\theta})$ of each $u$*, i.e. via functions $\hat{f} : \mathcal{G} \times \Theta \to \mathcal{Y}$ of the form $\hat{f}(G, \boldsymbol{\theta})$, where $\mathcal{G}$ is a space of architectures/computational graphs and $\Theta$ a space of parameters.[3] We are typically interested in learning *functionals* ($\mathcal{Y} \subseteq \mathbb{R}^d$) or *operators* ($\mathcal{Y} = \mathcal{G} \times \Theta$ or $\mathcal{Y} = \Theta$). To approximate the desired higher-order function, we assume access to a dataset of parameters sampled i.i.d. from an unknown distribution $p$ on $\mathcal{G} \times \Theta$. For example, in a supervised setup, we aim to optimise the following: $\operatorname{argmin}_{\hat{f} \in \mathcal{F}} \mathbb{E}_{(G,\boldsymbol{\theta}) \sim p} L\left(\mathfrak{F}(u_{G,\boldsymbol{\theta}}), \hat{f}(G, \boldsymbol{\theta})\right)$, where $L(\cdot, \cdot)$ is a loss function and $\mathcal{F}$ is an NN processing hypothesis class (e.g. metanetworks).

**Feedforward Neural Networks (FFNNs).** In this paper, we focus our analysis on Feedforward Neural Networks (FFNNs), i.e. linear layers interleaved with non-linearities. Consider NNs of the form $u_{G,\boldsymbol{\theta}} : \mathbb{R}^{d_{\text{in}}} \to \mathbb{R}^{d_{\text{out}}}$ of the following form:

$$\mathbf{x}_0 = \mathbf{x}, \quad \mathbf{x}_\ell = \sigma_\ell\left(\mathbf{W}_\ell \mathbf{x}_{\ell-1} + \mathbf{b}_\ell\right), \quad u_{G,\boldsymbol{\theta}}(\mathbf{x}) = \mathbf{x}_L \tag{1}$$

where $L$: the number of layers, $\mathbf{W}_i \in \mathbb{R}^{d_\ell \times d_{\ell-1}}$: the weights of the NN, $\mathbf{b}_i \in \mathbb{R}^{d_\ell}$: the biases of the NN, $d_0 = d_{\text{in}}$, $d_L = d_{\text{out}}$, $\sigma_\ell : \mathbb{R} \to \mathbb{R}$ activation functions applied element-wise. Here, the learnable parameters are $\boldsymbol{\theta} = (\mathbf{W}_1, \ldots, \mathbf{W}_L, \mathbf{b}_1, \ldots, \mathbf{b}_L)$ and the computational graph encodes the connections between vertices, *but also the type of activations used in each layer*.

**Neural Network symmetries.** One of the major difficulties with working with function parameters directly is that the same function can be represented with more than one parameter, i.e. there exists transformations that *when applied to any parameter $(G, \boldsymbol{\theta})$, keep the represented function intact*. Formally, an NN symmetry is induced by a set $\Psi$ of transformations $\psi : \mathcal{G} \times \Theta \to \mathcal{G} \times \Theta$, such that $u_{G,\boldsymbol{\theta}}(\mathbf{x}) = u_{\psi(G,\boldsymbol{\theta})}(\mathbf{x}), \forall \mathbf{x} \in \mathcal{X}, \forall (G, \boldsymbol{\theta}) \in \mathcal{G} \times \Theta$. If for a pair parameters $(G, \boldsymbol{\theta})$, $(G', \boldsymbol{\theta}')$, $\exists \psi$ such that $(G, \boldsymbol{\theta}) = \psi(G', \boldsymbol{\theta}')$, we will call the two parameters *equivalent* and write $(G, \boldsymbol{\theta}) \simeq (G', \boldsymbol{\theta}')$. To appropriately represent a *functional* $\mathfrak{F}$, a hypothesis (metanetwork) $\hat{f}$ should be *invariant* to transformations in $\Psi$: $\hat{f}(\psi(G, \boldsymbol{\theta})) = \hat{f}(G, \boldsymbol{\theta})$. For *operators*, $\hat{f}$ should be *equivariant* to transformations: $f(\psi(G, \boldsymbol{\theta})) = \psi(f(G, \boldsymbol{\theta}))$, such that identical functions map to identical functions.

**Permutation symmetries (connectionist structure).** For a fixed computational graph $G$, perhaps the most well-known symmetry of FFNNs are those induced by hidden neuron permutations [28]. *As far as metanetworks are concerned it is to date the only NN symmetry that has been accounted for* - see Section 2. This symmetry implies that permuting hidden neurons (along with their biases and incoming and outgoing weights) within each layer preserves the NN function (regardless of the activation function). This reads:

$$\mathbf{W}'_\ell = \mathbf{P}_\ell \mathbf{W}_\ell \mathbf{P}_{\ell-1}^{-1}, \ \mathbf{b}'_\ell = \mathbf{P}_\ell \mathbf{b}_\ell \implies (\mathbf{W}'_\ell, \mathbf{b}'_\ell)_{\ell=1}^L = \boldsymbol{\theta}' \simeq \boldsymbol{\theta} = (\mathbf{W}_\ell, \mathbf{b}_\ell)_{\ell=1}^L, \tag{2}$$

where $\ell \in \{1, \ldots, L\}$, $\mathbf{P}_0 = \mathbf{P}_L = \mathbf{I}$ and $\mathbf{P}_\ell \in \mathbb{R}^{d_\ell \times d_\ell}$ are arbitrary permutation matrices. Observe that they are different for each layer, with the input and output neurons held fixed.

**Graph Metanetworks (GMNs).** A recently introduced weight space architecture [33, 44], takes advantage of the permutation symmetries and treats FFNNs (among others, e.g. CNNs) as graphs, processing them with conventional GNNs. In particular, let $G = (\mathcal{V}, \mathcal{E})$ be the computational graph, $i \in \mathcal{V}$ an arbitrary vertex in the graph (neuron) and $(i, j) \in \mathcal{E}$ an arbitrary edge from vertex $j$ to vertex $i$.[4] Additionally, let $\mathbf{x}_V \in \mathbb{R}^{|\mathcal{V}| \times d_v}$ be the vertex features and $\mathbf{x}_E \in \mathbb{R}^{|\mathcal{E}| \times d_e}$ the edge features (i.e. biases and weights resp. in a FFNN). The general form of a $T$ iteration (layer) GMN reads:

$$\mathbf{h}_V^0(i) = \text{INIT}_V\left(\mathbf{x}_V(i)\right), \quad \mathbf{h}_E^0(i, j) = \text{INIT}_E\left(\mathbf{x}_E(i, j)\right) \tag{Init}$$

$$\mathbf{m}_V^t(i) = \bigoplus_{j \in \mathcal{N}(i)} \text{MSG}_V^t\left(\mathbf{h}_V^{t-1}(i), \mathbf{h}_V^{t-1}(j), \mathbf{h}_E^{t-1}(i, j)\right) \tag{Msg}$$

$$\mathbf{h}_V^t(i) = \text{UPD}_V^t\left(\mathbf{h}_V^{t-1}(i), \mathbf{m}_V^t(i)\right), \quad \mathbf{h}_E^t(i, j) = \text{UPD}_E^t\left(\mathbf{h}_V^{t-1}(i), \mathbf{h}_V^{t-1}(j), \mathbf{h}_E^{t-1}(i, j)\right) \tag{Upd}$$

$$\mathbf{h}_G = \text{READ}\left(\left\{\mathbf{h}_V^T(i)\right\}_{i \in \mathcal{V}}\right), \tag{Readout}$$

---

[3]Note the difference with architectures that access $u$ via input-output pairs $(\mathbf{x}_i, u_{G,\boldsymbol{\theta}}(\mathbf{x}_i))$ [46, 34].

[4]We use this convention to align with the indexing of the weights $\mathbf{W}(i, j)$.

where $\mathbf{h}_V^t, \mathbf{h}_E^t$ are vertex and edge representations at iteration $t$ and $\mathbf{h}_G$ is the overall graph (NN) representation. INIT, MSG, UPD are general function approximators (e.g. MLPs), while READ is a permutation invariant aggregator (e.g. DeepSets [81]). The above equations have appeared with several variations in the literature, e.g. in some cases the edge representations are not updated or the readout might involve edge representations as well. Another frequent strategy is to use *positional encodings* $\mathbf{p}_V, \mathbf{p}_E$ to break undesired symmetries. In FFNNs, Eq. (2) reveals that input and output vertices are not permutable, while vertices cannot be permuted across layers. Therefore, vertices (or edges) that are permutable share the same positional encoding (see Appendix A.1.2 for more details). **Remark:** Although, typically, the neighbourhood $\mathcal{N}(i)$ contains both incoming and outgoing edges, in Section 5 we will illustrate our method using only incoming edges: *forward neighbourhood* $\mathcal{N}_{\mathrm{FW}}(i) = \{j \in \mathcal{V} \mid \mathrm{layer}\,(i) - \mathrm{layer}\,(j) = 1\}$ and *backward* where $\mathrm{layer}\,(i)$ gives the layer neuron $i$ belongs. Backward neighbourhoods $\mathcal{N}_{\mathrm{BW}}(i)$ are defined defined similarly. In Appendix A.2, we show a more elaborate *bidirectional version* of our method, with both neighbourhoods considered.

## 4  Scaling symmetries in Feedforward Neural Networks

**Scaling symmetries (activation functions).** Intuitively, permutation symmetries stem from the *graph structure* of neural networks, or put differently, from the fact that hidden neurons do not possess any inherent ordering. Apart from the affine layers $\mathbf{W}_\ell$ that give rise to the graph structure, it is frequently the case that **activation functions** $\sigma_\ell$ have inherent symmetries that are bestowed to the NN.

Let us dive into certain illustrative examples: for the `ReLU` activation $\sigma(x) = \max(x, 0)$ it holds that $\sigma(ax) = \max(ax, 0) = a\max(x, 0), \forall a > 0$. For the `tanh` and `sine` activations $\sigma(x) = \tanh(x)$, $\sigma(x) = \sin(x)$ respectively, it holds that $\sigma(ax) = a\sigma(x), \forall a \in \{-1, 1\}$. In a slightly more complex example, polynomial activations $\sigma(x) = x^k$, we have $\sigma(ax) = a^d\sigma(x)$, i.e. the multiplier differs between input and output. In general, we will be talking about *scaling symmetries* whenever there exist pairs $(a, b)$ for which it holds that $\sigma(ax) = b\sigma(x)$. To see how such properties affect NN symmetries, let us focus on FFNNs (see Appendix A.3 for CNNs): for a neuron $i$ (we omit layer subscripts) we have $\sigma\big(a\mathbf{W}(i,:)\mathbf{x} + a\mathbf{b}(i)\big) = b\sigma\big(\mathbf{W}(i,:)\mathbf{x} + \mathbf{b}(i)\big)$, i.e. *multiplying its bias and all incoming weights with a constant $a$ results in scaling its output with a corresponding constant $b$.* Generalising this to linear transformations, we may ask the following: which are the pairs of matrices $(\mathbf{A}, \mathbf{B})$ for which we have $\sigma\big(\mathbf{A}\mathbf{W}\mathbf{x} + \mathbf{A}\mathbf{b}\big) = \mathbf{B}\sigma\big(\mathbf{W}\mathbf{x} + \mathbf{b}\big)$? Godfrey et al. [25] provide an answer for *any activation that respects certain conditions*. We restate here their most important results:

**Proposition 4.1** (Lemma 3.1. and Theorem E.14 from [25]). *Consider an activation function $\sigma : \mathbb{R} \to \mathbb{R}$. Under mild conditions,[5] the following hold:*

- *For any $d \in \mathbb{N}^+$, there exists a (non-empty) group of invertible matrices defined as: $I_{\sigma,d} = \{\mathbf{A} \in \mathbb{R}^{d\times d} : \text{invertible} \mid \exists\, \mathbf{B} \in \mathbb{R}^{d\times d} \text{ invertible, such that: } \sigma(\mathbf{A}\mathbf{x}) = \mathbf{B}\sigma(\mathbf{x})\}$ (**intertwiner group**), and a mapping function $\phi_{\sigma,d}$ such that $\mathbf{B} = \phi_{\sigma,d}(\mathbf{A})$.*

- *Every $\mathbf{A} \in I_{\sigma,d}$ is of the form $\mathbf{PQ}$, where $\mathbf{P}$: permutation matrix and $\mathbf{Q} = \mathrm{diag}\big(q_1, \ldots q_d\big)$ diagonal, with $q_i \in D_\sigma = \{a \in \mathbb{R} \setminus \{0\} \mid \sigma(ax) = \phi_{\sigma,1}(a)\sigma(x)\}$: the 1-dimensional group, and $\phi_{\sigma,d}(\mathbf{A}) = \mathbf{P}\mathrm{diag}\big(\phi_{\sigma,1}(q_1), \ldots \phi_{\sigma,1}(q_d)\big)$.*

This is a powerful result that completely answers the question above for most practical activation functions. Importantly, not only does it recover permutation symmetries, but also reveals symmetries to diagonal matrix groups, which can be identified by solely examining $\phi_{\sigma,1}$, i.e. the one-dimensional case and the set $D_\sigma$ (easily proved to be a group) we have already discussed in our examples above.

Using this statement, Godfrey et al. [25] characterised various activation functions (or recovered existing results), e.g. ReLU: $I_{\sigma,d}$ contains **generalised permutation matrices with positive entries** of the form $\mathbf{PQ}$, $\mathbf{Q} = \mathrm{diag}(q_1, \ldots, q_d)$, $q_i > 0$ and $\phi_{\sigma,d}(\mathbf{PQ}) = \mathbf{PQ}$ [56]. Additionally, here we characterise the intertwiner group of `sine` (used in the popular SIREN architecture [70] for INRs). Not surprisingly, it has the same intertwiner group with `tanh` [11, 21] (we also recover this here using Proposition 4.1). Formally, (proof in Appendix A.7.1):

**Corollary 4.2.** *Hyperbolic tangent $\sigma(x) = \tanh(x)$ and sine activation $\sigma(x) = \sin(\omega x)$, satisfy the conditions of Proposition 4.1, when (for the latter) $\omega \neq k\pi, k \in \mathbb{Z}$. Additionally, $I_{\sigma,d}$ contains **signed permutation matrices** of the form $\mathbf{PQ}$, with $\mathbf{Q} = \mathrm{diag}(q_1, \ldots, q_d)$, $q_i = \pm 1$ and $\phi_{\sigma_d}(\mathbf{PQ}) = \mathbf{PQ}$.*

---

[5] See Appendix A.7.1 for the precise statement and more details about $\phi_{\sigma,d}$.

It is straightforward to see that the symmetries of Proposition 4.1, induce equivalent parameterisations for FNNNs. In particular, it follows directly from Proposition 3.4. in [25], that for activation functions $\sigma_\ell$ satisfying the conditions of Proposition 4.1 and when $\phi_{\sigma,\ell}(\mathbf{Q}) = \mathbf{Q}$, we have that:

$$\mathbf{W}'_\ell = \mathbf{P}_\ell \mathbf{Q}_\ell \mathbf{W}_\ell \mathbf{Q}_{\ell-1}^{-1} \mathbf{P}_{\ell-1}^{-1}, \ \mathbf{b}'_\ell = \mathbf{P}_\ell \mathbf{Q}_\ell \mathbf{b}_\ell \implies (\mathbf{W}'_\ell, \mathbf{b}'_\ell)_{\ell=1}^L = \boldsymbol{\theta}' \simeq \boldsymbol{\theta} = (\mathbf{W}_\ell, \mathbf{b}_\ell)_{\ell=1}^L, \quad (3)$$

where again $\ell \in \{1, \ldots, L\}, \mathbf{P}_0 = \mathbf{Q}_0 = \mathbf{P}_L = \mathbf{Q}_L = \mathbf{I}$.

## 5 Scale Equivariant Graph MetaNetworks

As previously mentioned, the metanetworks operating on weight spaces that have been proposed so far, either do not take any symmetries into account or are invariant/equivariant to permutations alone as dictated by Eq. (2). In the following section, we introduce an architecture invariant/equivariant to **permutations and scalings**, adhering to Eq. (3). An important motivation for this is that in various setups, *these are the only function-preserving symmetries*, i.e. for a fixed graph $u_{G,\boldsymbol{\theta}} = u_{G,\boldsymbol{\theta}'} \Rightarrow (\boldsymbol{\theta}, \boldsymbol{\theta}')$ satisfy Eq. (3) - e.g. see [21] for the conditions for `tanh` and [61, 26] for `ReLU`.

**Main idea.** Our framework is similar in spirit to most works on equivariant and invariant NNs [10]. In particular, we build equivariant GMNs that will preserve both symmetries at vertex- and edge-level, i.e. *vertex representations will have the same symmetries with the biases and edge representations with the weights*. To see this, suppose two parameter vectors are equivalent according to Eq. (3). Then, the *hidden* neurons representations - the discussion on *input/output* neurons is postponed until Appendix A.1.4 - should respect the following (the GMN iteration $t$ is omitted to simplify notation):

$$\mathbf{h}'_V(i) = q_\ell(\pi_\ell(i)) \mathbf{h}_V(\pi_\ell(i)), \quad \ell = \text{layer}(i) \in \{1, \ldots, L-1\} \quad (4)$$

$$\mathbf{h}'_E(i,j) = q_\ell(\pi_\ell(i)) \mathbf{h}_E(\pi_\ell(i), \pi_{\ell-1}(j)) q_{\ell-1}^{-1}(\pi_{\ell-1}(j)), \ \ell = \text{layer}(i) \in \{2, \ldots, L-1\}, \quad (5)$$

where $\pi_\ell : \mathcal{V}_\ell \leftrightarrow \mathcal{V}_\ell$ permutes the vertices of layer $\ell$ (denoted with $\mathcal{V}_\ell$) according to $\mathbf{P}_\ell$ and $q_\ell : \mathcal{V}_\ell \to \mathbb{R} \setminus \{0\}$ scales the vertex representations of layer $\ell$ according to $\mathbf{Q}_\ell$. We will refer to the latter as *forward scaling* in Eq. (4) and *bidirectonal scaling* in Eq. (5). To approximate *operators* (equivariance), we compose multiple equivariant GMN layers/iterations and in the end, project vertex/edge representations to the original NN weight space, while to approximate *functionals* (invariance), we compose a final invariant one in the end summarising the input to a scalar/vector.

To ease exposition, we will first discuss our approach w.r.t. vertex representations. Assume that vertex representation symmetries are preserved by the initialisation of the MPNN - Eq. (Init) - and so are edge representation symmetries for all MPNN layers. Therefore, we can only focus on the message passing and vertex update steps - Eq. (Msg) and Eq. (Upd). Additionally, let us first focus on hidden neurons and assume only forward neighbourhoods. The following challenges arise:

**Challenge 1 - Scale Invariance / Equivariance.** First off, the message and the update function $\text{MSG}_V, \text{UPD}_V$ should be *equivariant to scaling* - in this case to the forward scaling using the multiplier of the central vertex $q_\ell(i)$. Additionally, the readout READ, apart from being permutation invariant should also be *invariant to the different scalar multipliers of each vertex*. Dealing with this requires devising functions of the following form:

$$g_i(q_1 \mathbf{x}_1, \ldots, q_n \mathbf{x}_n) = q_i g_i(\mathbf{x}_1, \ldots, \mathbf{x}_n), \forall q_i \in D_i, i \in \{1, \ldots, n\} \quad (6)$$

where $D_i$ a 1-dimensional scaling group as defined in Proposition 4.1. Common examples are those discussed in Section 4, e.g. $D_i = \{1, -1\}$ **or** $D_i = \mathbb{R}^+$. The first case, i.e. *sign symmetries*, has been discussed in recent work [43, 40]. Here we generalise their architecture into arbitrary scaling groups. In specific, *Scale Equivariant* networks follow the methodology of [40], i.e. they are compositions of multiple linear transformations multiplied elementwise with the output of *Scale Invariant* functions:

$$\text{ScaleInv}^k(\mathbf{X}) = \rho^k(\tilde{\mathbf{x}}_1, \ldots, \tilde{\mathbf{x}}_n), \quad \text{(Scale Inv. Net)}$$

$$\text{ScaleEq} = \mathsf{f}^K \circ \cdots \circ \mathsf{f}^1, \ \mathsf{f}^k(\mathbf{X}) = (\boldsymbol{\Gamma}_1^k \mathbf{x}_1, \ldots, \boldsymbol{\Gamma}_n^k \mathbf{x}_n) \odot \text{ScaleInv}^k(\mathbf{X}), \quad \text{(Scale Equiv. Net)}$$

where $\rho^k : \prod_{i=1}^n \mathcal{X}_i \to \mathbb{R}^{\sum_{i=1}^n d_i^k}$ universal approximators (e.g. MLPs), $\boldsymbol{\Gamma}_i^k : \mathcal{X}_i \to \mathbb{R}^{d_i^k}$ linear transforms (for each of the $k$ invariant/equivariant layers resp - in practice, we observed experimentally that a single layer $K = 1$ was sufficient), and $\tilde{\mathbf{x}}_i$ are explained in detail in the following paragraph.

Central to our method is defining a way to achieve invariance. One option is *canonicalisation*, i.e. by defining a function canon $: \mathcal{X} \to \mathcal{X}$ *that maps all equivalent vectors* $\mathbf{x}$ *to a representative* (obviously

non-equivalent vectors have different representatives): Iff $\mathbf{x} \simeq \mathbf{y} \in \mathcal{X}$, then $\mathbf{x} \simeq \mathsf{canon}(\mathbf{x}) = \mathsf{canon}(\mathbf{y})$. In certain cases, these are easy to define and *differentiable almost everywhere*, for example for positive scaling: $\mathsf{canon}(\mathbf{x}) = \frac{\mathbf{x}}{\|\mathbf{x}\|}$.[6] For sign symmetries, this is not as straightforward: in 1-dimension one can use $|x|$, but for arbitrary dimensions, a more complex procedure is required, as recently discussed in [49, 48]. Since the group is small - two elements - one can use *symmetrisation* instead [79], as done by Lim et al. [41]: $\mathsf{symm}(\mathbf{x}) = \sum_{\mathbf{y}:\mathbf{y}\simeq\mathbf{x}} \mathsf{MLP}(\mathbf{y})$, i.e. for sign: $\mathsf{symm}(\mathbf{x}) = \mathsf{MLP}(\mathbf{x}) + \mathsf{MLP}(-\mathbf{x})$. Therefore, we define: $\mathsf{ScaleInv}^k(\mathbf{X}) = \rho^k(\tilde{\mathbf{x}}_1, \ldots, \tilde{\mathbf{x}}_n)$, with $\tilde{\mathbf{x}}_i = \mathsf{canon}(\mathbf{x}_i)$ or $\tilde{\mathbf{x}}_i = \mathsf{symm}(\mathbf{x}_i)$. Importantly, it is known that both cases *allow for universality*, see [8, 29] and [79, 62] respectively.

**Challenge 2 - Rescaling: Different multipliers.** Scale equivariance alone is not sufficient, since the input vectors of the message function $\mathsf{MSG}_V$ are *scaled by different multipliers* - central vertex: $q_\ell(i)$, neighbour: $q_{\ell-1}(j)$, edge: $q_\ell(i)q_{\ell-1}^{-1}(j)$, *while its output should be scaled differently as well* - $q_\ell(i)$. We refer to this problem as *rescaling*. Dealing with Challenge 2 requires functions of the form:

$$g\big(q_1\mathbf{x}_1, \ldots q_n\mathbf{x}_n\big) = g(\mathbf{x}_1, \ldots \mathbf{x}_n) \prod_{i=1}^{n} q_i, \forall q_i \in D_i. \tag{7}$$

We call these functions *rescale equivariant*. Note, that this is an unusual symmetry in equivariant NN design. Our approach is based on the observation that *any n-other monomial containing variables from all vectors $\mathbf{x}_i$ is rescale-equivariant*. Collecting all these monomials into a single representation is precisely the *outer product* $\mathbf{X}_n = \mathbf{x}_1 \otimes \cdots \otimes \mathbf{x}_n$, where $\mathbf{X}_n(j_1, \ldots, j_n) = \prod_{i=1}^{n} \mathbf{x}_i(j_i)$. Therefore, the general form of our proposed Rescale Equivariant Network is as follows:

$$\mathsf{ReScaleEq}(\mathbf{x}_1, \ldots \mathbf{x}_n) = \mathsf{ScaleEq}\big(\mathsf{vec}(\mathbf{X}_n)\big). \tag{ReScale Equiv. Net}$$

In practice, given that the size of $\mathbf{X}_n$ grows polynomially with $n$, we resort to a more computationally friendly subcase, i.e. *hadamard products*, i.e. $\mathsf{ReScaleEq}(\mathbf{x}_1, \ldots \mathbf{x}_n) = \odot_{i=1}^{n} \mathbf{\Gamma}_i \mathbf{x}_i, \mathbf{\Gamma}_i : \mathcal{X}_i \to \mathbb{R}^d$. Contrary to the original formulation, the latter is linear (lack of multiplication with an invariant layer).

**Scale Equivariant Message Passing.** We are now ready to define our message-passing scheme. Starting with the message function, we require each message vector $\mathbf{m}_V(i)$ to have the same symmetries as the central vertex $i$. Given the scaling symmetries of the neighbour and the edge, for forward neighbourhoods, this reads: $\mathsf{MSG}_V\big(q_x\mathbf{x}, q_y\mathbf{y}, q_x q_y^{-1}\mathbf{e}\big) = q_x\mathsf{MSG}_V(\mathbf{x}, \mathbf{y}, \mathbf{e})$. In this case, we opt to eliminate $q_y$ by multiplication as follows:

$$\mathsf{MSG}_V(\mathbf{x}, \mathbf{y}, \mathbf{e}) = \mathsf{ScaleEq}\left([\mathbf{x}, \mathsf{ReScaleEq}(\mathbf{y}, \mathbf{e})]\right), \tag{8}$$

where $[\cdot, \cdot]$ denotes concatenation, $\mathsf{ReScaleEq}(q_y\mathbf{y}, q_x q_y^{-1}\mathbf{e}) = q_x\mathsf{ReScaleEq}(\mathbf{y}, \mathbf{e})$. In our experiments, we used only $\mathbf{y}$ and $\mathbf{e}$, since we did not observe significant performance gains by including the central vertex $\mathbf{x}$. Now, the update function is straightforward to implement since it receives vectors with the same symmetry, i.e. it should hold that: $\mathsf{UPD}_V(q_x\mathbf{x}, q_x\mathbf{m}) = q_x\mathsf{UPD}_V(\mathbf{x}, \mathbf{m})$ which is straightforward to implement with a scale equivariant network, after concatenating $\mathbf{x}$ and $\mathbf{m}$:

$$\mathsf{UPD}_V(\mathbf{x}, \mathbf{m}) = \mathsf{ScaleEq}\left([\mathbf{x}, \mathbf{m}]\right). \tag{9}$$

Finally, to summarise our graph into a single scalar/vector we require a scale and permutation-invariant readout. The former is once more achieved using canonicalised/symmetrised versions of the vertex representations of hidden neurons, while the latter using a DeepSets architecture as usual:

$$\mathsf{READ}_V(\mathbf{X}) := \mathsf{DeepSets}(\tilde{\mathbf{x}}_1, \ldots, \tilde{\mathbf{x}}_n), \quad \tilde{\mathbf{x}}_i = \mathsf{canon}_i(\mathbf{x}_i) \text{ or } \tilde{\mathbf{x}}_i = \mathsf{symm}_i(\mathbf{x}_i) \tag{10}$$

In Appendix A.1, we show how to handle symmetries in the rest of the architecture components (i.e. *initialisation*, *positional encodings*, *edge updates and i/o vertices*) and provide an extension of our method to *bidirectional message-passing* (Appendix A.2), which includes backward neighbourhoods.

**Expressive power.** Throughout this section, we discussed only scaling symmetries and not permutation symmetries. However, it is straightforward to conclude that ScaleGMN is also *permutation equivariant/invariant*, since it is a subcase of GMNs; if one uses universal approximators in their message/update functions (MLP), our corresponding functions will be expressible by this architecture, which was proved to be permutation equivariant in [44]. Although this implies that GMN can express ScaleGMN, this is expected since $\mathbf{PQ}$ symmetries are more restrictive than just $\mathbf{P}$. Note that these symmetries are always present in FFNNs, and thus it is desired to explicitly model them, to introduce a more powerful *inductive bias*. Formally, on symmetry preservation (proved in Appendix A.7.2):

---

[6]Fwd: $\mathbf{x} \simeq \mathbf{y} \Rightarrow \mathbf{x} = q\mathbf{y} \Rightarrow \frac{\mathbf{y}}{\|\mathbf{y}\|} = \frac{q\mathbf{x}}{|q|\|\mathbf{x}\|} = \frac{\mathbf{x}}{\|\mathbf{x}\|}$ ($q > 0$). Reverse: $\frac{\mathbf{x}}{\|\mathbf{x}\|} = \frac{\mathbf{y}}{\|\mathbf{y}\|} \Rightarrow \mathbf{x} = \frac{\|\mathbf{x}\|}{\|\mathbf{y}\|}\mathbf{y} \Rightarrow \mathbf{x} \simeq \mathbf{y}$

Table 1: INR classification on MNIST, F-MNIST, CIFAR-10 and Aug. CIFAR-10. We train all methods on 3 seeds and report the mean and std. (*) denotes the baselines trained by us and we report the rest as in the corresponding papers. Colours denote **First**, Second and Third.

| Method | MNIST | F-MNIST | CIFAR-10 | Augmented CIFAR-10 |
|---|---|---|---|---|
| MLP | $17.55 \pm 0.01$ | $19.91 \pm 0.47$ | $11.38 \pm 0.34$* | $16.90 \pm 0.25$ |
| Inr2Vec [47] | $23.69 \pm 0.10$ | $22.33 \pm 0.41$ | - | - |
| DWS [54] | $85.71 \pm 0.57$ | $67.06 \pm 0.29$ | $34.45 \pm 0.42$ | $41.27 \pm 0.026$ |
| NFN$_{NP}$ [85] | $78.50 \pm 0.23$* | $68.19 \pm 0.28$* | $33.41 \pm 0.01$* | $46.60 \pm 0.07$ |
| NFN$_{HNP}$ [85] | $79.11 \pm 0.84$* | $68.94 \pm 0.64$* | $28.64 \pm 0.07$* | $44.10 \pm 0.47$ |
| NG-GNN [33] | $91.40 \pm 0.60$ | $68.00 \pm 0.20$ | $36.04 \pm 0.44$* | $45.70 \pm 0.20$* |
| ScaleGMN (Ours) | $96.57 \pm 0.10$ | $80.46 \pm 0.32$ | $36.43 \pm 0.41$ | $56.62 \pm 0.24$ |
| ScaleGMN-B (Ours) | $\mathbf{96.59 \pm 0.24}$ | $\mathbf{80.78 \pm 0.16}$ | $\mathbf{38.82 \pm 0.10}$ | $\mathbf{56.95 \pm 0.57}$ |

Table 2: Generalization pred.: Kendall-$\tau$ on subsets of SmallCNN Zoo w/ ReLU/Tanh activations.

| Method | CIFAR-10-GS ReLU | SVHN-GS ReLU | CIFAR-10-GS Tanh | SVHN-GS Tanh | CIFAR-10-GS both act. |
|---|---|---|---|---|---|
| StatNN [74] | $0.9140 \pm 0.001$ | $0.8463 \pm 0.004$ | $0.9140 \pm 0.000$ | $0.8440 \pm 0.001$ | $0.915 \pm 0.002$ |
| NFN$_{NP}$ [85] | $0.9190 \pm 0.010$ | $0.8586 \pm 0.003$ | $0.9251 \pm 0.001$ | $0.8580 \pm 0.004$ | $0.922 \pm 0.001$ |
| NFN$_{HNP}$ [85] | $0.9270 \pm 0.001$ | $0.8636 \pm 0.002$ | $0.9339 \pm 0.000$ | $0.8586 \pm 0.004$ | $0.934 \pm 0.001$ |
| NG-GNN [33] | $0.9010 \pm 0.060$ | $0.8549 \pm 0.002$ | $0.9340 \pm 0.001$ | $0.8620 \pm 0.003$ | $0.931 \pm 0.002$ |
| ScaleGMN (Ours) | $0.9276 \pm 0.002$ | $\mathbf{0.8689 \pm 0.003}$ | $0.9418 \pm 0.005$ | $\mathbf{0.8736 \pm 0.003}$ | $0.941 \pm 0.006$ |
| ScaleGMN-B (Ours) | $\mathbf{0.9282 \pm 0.003}$ | $0.8651 \pm 0.001$ | $\mathbf{0.9425 \pm 0.004}$ | $0.8655 \pm 0.004$ | $\mathbf{0.941 \pm 0.000}$ |

**Proposition 5.1.** *ScaleGMN is permutation & scale **equivariant**. Additionally, ScaleGMN is permutation & scale **invariant** when using a readout with the same symmetries.*

Finally, we analysed the ability of ScaleGMN to evaluate the input FFNN and its gradients, i.e. *simulate the forward and the backward pass of an input FFNN*. To see why this is a desired inductive bias, recall that a functional/operator can often be written as a function of input-output pairs (e.g. via an integral on the entire domain) or of the input function's derivatives (e.g. via a differential equation). By simulating the FFNN, one can reconstruct function evaluations and gradients, which an additional module can later combine. Formally (proof and precise statement in Appendix A.7.2):

**Theorem 5.2.** *Consider an FFNN as per Eq. (1) with activation functions respecting the conditions of Proposition 4.1. Assume a Bidirectional-ScaleGMN with sufficiently expressive message and vertex update functions. Then, ScaleGMN can simulate both the forward and the backward pass of the FFNN for arbitrary inputs, when ScaleGMN's iterations (depth) are $L$ and $2L$ respectively.*

## 6 Experiments

**Datasets.** We evaluate ScaleGMN on datasets containing NNs with three popular activation functions: `sine, ReLU` and `tanh`. The former is prevalent in INRs, which in turn are the most appropriate testbed for GMNs. We experiment with the tasks of *(1) INR classification* (invariant task), i.e. classifying functions (signals) represented as INRs and *(2) INR editing* (equivariant), i.e. transforming those functions. Additionally, `ReLU` and `tanh` are common in neural classifiers/regressors. A popular GMN benchmark for those is *(3) generalisation prediction* (invariant), i.e. predicting a classifier's test accuracy. Here, classifiers instead of FFNNs are typically CNNs (for computer vision tasks) and to this end, we extend our method to the latter in Appendix A.3. We use existing datasets that have been constructed by the authors of relevant methods, are publicly released and have been repeatedly used in the literature (8 datasets in total, 4 for each task). Finally, we follow established protocols: we perform a hyperparameter search and use the best-achieved validation metric throughout training to select our final model. We report the test metric on the iteration where the best validation is achieved.

Table 3: Dilating MNIST INRs. MSE between the reconstructed and ground-truth image.

| Method | MSE in $10^{-2}$ |
|---|---|
| MLP | $5.35 \pm 0.00$ |
| DWS [54] | $2.58 \pm 0.00$ |
| NFN$_{NP}$ [85] | $2.55 \pm 0.00$ |
| NFN$_{HNP}$ [85] | $2.65 \pm 0.01$ |
| NG-GNN-0 [33] | $2.38 \pm 0.02$ |
| NG-GNN-64 [33] | $2.06 \pm 0.01$ |
| ScaleGMN (Ours) | $2.56 \pm 0.03$ |
| ScaleGMN-B (Ours) | $\mathbf{1.89 \pm 0.00}$ |

Table 4: Ablation: Permutation equivariant models + scaling augmentations.

| Method | F-MNIST | CIFAR-10-GS ReLU |
|---|---|---|
| DWS [54] | $67.06 \pm 0.29$ | — |
| DWS [54] + aug. | $71.42 \pm 0.45$ | — |
| NFN$_{NP}$ [85] | $68.19 \pm 0.28$ | $0.9190 \pm 0.010$ |
| NFN$_{NP}$ [85] + aug. | $70.34 \pm 0.13$ | $0.8474 \pm 0.01$ |
| NFN$_{HNP}$ [85] | $68.94 \pm 0.64$ | $0.9270 \pm 0.001$ |
| NFN$_{HNP}$ [85] + aug. | $70.24 \pm 0.47$ | $0.8906 \pm 0.01$ |
| NG-GNN [33] | $68.0 \pm 0.20$ | $0.9010 \pm 0.060$ |
| NG-GNN [33] + aug. | $72.01 \pm 1.4$ | $0.8855 \pm 0.02$ |
| ScaleGMN (Ours) | $80.46 \pm 0.32$ | $0.9276 \pm 0.002$ |
| ScaleGMN-B (Ours) | $80.78 \pm 0.16$ | $0.9282 \pm 0.003$ |

**Baselines.** DWS [54] is a *non-local* metanetwork that uses all linear permutation equivariant/invariant layers interleaved with non-linearities. NFN$_{HNP}$ [85] is mathematically equivalent to DWS, while NFN$_{NP}$ [85] makes stronger symmetry assumptions in favour of parameter efficiency. The two variants are also designed to process CNNs contrary to DWS. NG-GNN [33] converts each input NN to a graph (similar to our work) and employs a GNN (in specific PNA [15]). Importantly, the last three methods transform the input parameters with random Fourier features while all methods perform input normalisations to improve performance and facilitate training. These tricks are in general not equivariant to scaling and were unnecessary in the ScaleGMN experiments (more details in Appendix A.4). On INR classification we include a naive MLP on the flattened parameter vector and Inr2Vec [47], a task-specific non-equivariant method. For generalization prediction, we also compare to StatNN [74], which predicts NN accuracy based on statistical features of its weights/biases.

**INR classification.** We design a ScaleGMN with permutation & *sign* equivariant components (and invariant readout). Additionally, we experiment with the bidirectional version denoted as *ScaleGMN-B*. We use the following datasets of increasing difficulty: *MNIST INR, F-MNIST INR* (grayscale images) and *CIFAR10-INR*, *Aug. CIFAR10-INR.*: INRs representing images from the MNIST [38], FashionMNIST [76] and CIFAR [35] datasets resp. and an augmented version of CIFAR10-INR containing 20 different INRs for each image, trained from different initialisations (often called *views*). The reported metric is *test accuracy*. Note that [85, 86] use only the augmented dataset and hence we rerun all baselines for the original version. All input NNs correspond to functions $u_{G,\theta} : \mathbb{R}^2 \to \mathbb{R}^c$, i.e. pixel coordinate to GS/RGB value. Further implementation details can be found in Appendix A.4.1. As shown in Table 1, *ScaleGMN consistently outperforms all baselines in all datasets considered*, with performance improvements compared to the state-of-the-art ranging from approx. 3% (CIFAR-10) to 13% (F-MNIST). While previous methods often resort to additional engineering strategies, such as probe features and advanced architectures [33] or extra training samples [85] to boost performance, in our case this was possible using vanilla ScaleGMNs. For example, in MNIST and F-MNIST, NG-GNN achieves $94.7 \pm 0.3$ and $74.2 \pm 0.4$ with 64 probe features which is apprpx. 2% and 6% below our best performance. The corresponding results for NG-T [33] (transformer) were $97.3 \pm 0.2$ and $74.8 \pm 0.9$, still on par or approx. 6% below ScaleGMN. Note that all the above are orthogonal to our work and can be used to further improve performance.

**Predicting CNN Generalization from weights.** As in prior works. we consider datasets of image classifiers and measure predictive performance using Kendall's $\tau$ [30]. We select the two datasets used in [85], namely CIFAR-10-GS and SVHN-GS originally from *Small CNN Zoo* [74]. These contain CNNs with `ReLU` or `tanh` activations, which exhibit scale and sign symmetries respectively. To assess the performance of our method (1) on each activation function individually and (2) on a dataset with heterogeneous activation functions we discern distinct paths for this experiment. In the first case, we split each dataset into two subsets each containing the same activation and evaluate all baselines. As shown in Table 2, once again *ScaleGMN outperforms all the baselines in all the examined datasets*. This highlights the ability of our method to be used across different activation functions and architectures. Performance improvements here are less prominent due to the hardness of the task, a phenomenon also observed in the comparisons between prior works. Note that in this case, additional symmetries arise by the softmax classifier [36], which are currently not accounted for by none of the methods. We additionally evaluate our method on *heterogeneous activation functions*.

In principle, our method does *not* impose limitations regarding the homogeneity of the activation functions of the input NNs - all one needs is to have a different canonicalisation/symmetrisation module for each activation. Experiments on CIFAR-10-GS show that ScaleGMN yields superior performance compared to the baselines, significantly exceeding the performance of the next-best model. Further implementation details can be found in Appendix A.4.2.

**INR editing.** Here, our goal is to transform the weights of the input NN, to modify the underlying signal to a new one. In this case, our method should be *equivariant* to the permutation and scaling symmetries, such that every pair of equivalent input NNs is mapped to a pair of equivalent output NNs. Hence, we employ a ScaleGMN similar to the above experiments but omit the invariant readout layer. Following [33], we evaluate our method on the MNIST dataset and train our model to dilate the encoded MNIST digits. Further implementation details can be found in Appendix A.4.3. As shown in Table 3, our bidirectional ScaleGMN-B achieves an MSE test loss ($10^{-2}$) equal to $1.891$, *surpassing all permutation equivariant baselines*. Notably, our method also outperforms the NG-GNN [33] baseline that uses 64 probe features. Additionally, our forward variant, ScaleGMN, performs on par with the previous permutation equivariant baselines with an MSE loss of $2.56$. Note that the performance gap between the forward and the bidirectional model is probably expected for equivariant tasks: here we are required to compute representations for every node of the graph, yet in the forward variant, the earlier the layer of the node, the smaller the amount of information it receives. This observation corroborates the design choices of the baselines, which utilize either bidirectional mechanisms (NG-GNN [33]) or non-local operations (NFN [85]).

**Ablation study: Scaling data augmentations.** We baseline our method with permutation equivariant methods trained with scaling augmentations: For every training datapoint, at each training iteration, we sample a diagonal scaling matrix for every hidden layer of the input NN and multiply it with the weights/bias matrices as per Eq. (3) (omitting the permutation matrices). We sample the elements of the matrices independently as follows: *Sign symmetry:* Bernoulli distribution with probability 0.5. *Positive scaling*: Exponential distribution where the coefficient $\lambda$ is a hyperparameter that we tune on the validation set. Observe here that designing augmentations in the latter case is a particularly challenging task since we have to sample from a *continuous* and *unbounded* distribution. Our choice of the exponential was done by consulting the norm plots where in some cases the histogram resembles an exponential Fig. 1. Nevertheless, regardless of the distribution choice we cannot guarantee that the augmentations will be sufficient to achieve (approximate) scaling equivariance, due to the lack of upper bound. We evaluate on the F-MNIST dataset for the sign symmetry and on the CIFAR-10-GS-ReLU for the positive scale. As shown in Table 4, regarding the former, augmenting the training set leads consistently to better results when compared to the original baselines. *None of these methods however achieved results on par with ScaleGMN and ScaleGMN-B. On the other hand, we were unable to even surpass the original baselines regarding the latter task*. This indicates that designing an effective positive scaling augmentation might be a non-trivial task.

**Limitations.** A limitation of our work is that it is currently designed for FFNNs and CNNs and does not cover other layers that either modify the computational graph (e.g. skip connections) or introduce additional symmetries (e.g. softmax and normalisation layers). In both cases, in future work, we plan to characterise scaling symmetries (certain steps were made in [25]) and modify ScaleGMN for general computational graphs as in [44, 88]. Additionally, a complete characterisation of the functions that can be expressed by our scale/rescale equivariant building blocks is an open question (except for sign [40]). Finally, an important theoretical matter is a complete characterisation of the expressive power of ScaleGMN, similar to all equivariant metanetworks.

# 7  Conclusion

In this work, we propose ScaleGMN  a metanetwork framework that introduces to the field of NN processing a stronger inductive bias: accounting for function-preserving scaling symmetries that arise from activation functions. ScaleGMN can be applied to NNs with various activation functions by modifying any graph metanetwork, is proven to enjoy desirable theoretical guarantees and empirically demonstrates the significance of scaling by improving the state-of-the-art in several datasets. With our work, we aspire to introduce a new research direction, i.e. incorporating into metanetworks various NN symmetries beyond permutations, aiming to improve their generalisation capabilities and broaden their applicability in various NN processing domains.

## Acknowledgements

IK, GB and YP were partially supported by project MIS 5154714 of the National Recovery and Resilience Plan Greece 2.0 funded by the European Union under the NextGenerationEU Program. This work was partially supported by a computational resources grant from The Cyprus Institute (HPC system "Cyclone") as well as from an AWS credits grant provided by GRNET – National Infrastructures for Research and Technology.

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

# A Appendix / supplemental material

## A.1 Additional practical considerations of ScaleGMN

In the following section, we describe the additional practical considerations that were omitted from Section 5. For the purposes of this section, we introduce the functions $\text{layer}(i)$ and $\text{pos}(i)$ which return the layer of a vertex $i$ and its position within this layer respectively.

### A.1.1 Initialisation

As previously discussed, all vertex/edge representations throughout the network should respect the symmetries given by Eqs. (4), (5). This is straightforward to achieve at initialisation, by using the NN biases and weights (similarly to [33]). It is possible to use a scale equivariant layer here as well, e.g. a linear layer, which is what was performed in our empirical evaluation. For notation convenience, denote the biases and weights of Eq. (1), with $\tilde{\mathbf{b}}, \tilde{\mathbf{W}}$, such that $\tilde{\mathbf{b}}(i) = \mathbf{b}_{\text{layer}(i)}(\text{pos}(i))$, $\tilde{\mathbf{W}}(i, j) = \mathbf{W}_{\text{layer}(i)}(\text{pos}(i), \text{pos}(j))$. Now we can define the initialisation:

$$\text{INIT}_V\left(\mathbf{x}_V(i)\right) = \boldsymbol{\Gamma}_V \mathbf{x}_V(i), \quad \mathbf{x}_V(i) = \begin{cases} \tilde{\mathbf{b}}(i), & \text{if layer}(i) \in \{1, \ldots, L\}, \\ 1, & \text{if layer}(i) = 0. \end{cases} \tag{11}$$

$$\text{INIT}_E\left(\mathbf{x}_E(i, j)\right) = \boldsymbol{\Gamma}_E \mathbf{x}_E(i, j), \quad \mathbf{x}_E(i, j) = \tilde{\mathbf{W}}(i, j), \text{ if layer}(i) \in \{1, \ldots, L\}. \tag{12}$$

Note that the above is different from the initialisation/graph construction used in [44]. For example, in this work, an extra vertex is created for each bias, which is connected to its corresponding neuron vertex with an edge initialised with the bias value. However, this strategy would complicate scaling symmetry preservation and was thus not used in our work.

### A.1.2 Positional Encodings

As discussed in Section 3, *not all vertices/edges of an FFNN are permutable*. In particular, vertices corresponding to the input and output neurons are fixed, while vertices corresponding to hidden neurons can be permuted only within the same layer. Similarly, edges originating from an input neuron are permutable only with edges originating from the same neuron, and the same holds for edges terminating at an output neuron, while edges of hidden layers can be only permuted within the same layer.

To adhere to the above, we need to devise a *symmetry-breaking* mechanism, to ensure that the GMN is **not** equivariant to undesired symmetries (e.g. permuting output neurons). This is the role of positional encodings, which are learnable vectors (in the literature fixed vectors have been also used), shared across all input graphs, that take part in message-passing. In particular, for vertex positional encodings the following holds:

$$\mathbf{p}_V(i) = \mathbf{p}_V(j), \text{ if layer}(i) = \text{layer}(j) \neq \{0, L\}, \tag{13}$$

and for edge positional encodings:

$$\mathbf{p}_E(i, j) = \mathbf{p}_E(i', j'), \text{ if } \begin{cases} j = j', \text{layer}(j) \in \{0, L\}, \\ \text{layer}(j) = \text{layer}(j') \neq 0, \text{layer}(i) = \text{layer}(i') \neq L, \\ i = i', \text{layer}(i) \in \{0, L\}, \end{cases} \tag{14}$$

while otherwise, the positional encodings are free to take different values. Symmetry-breaking usually happens at initialisation (which is then inherited by the subsequent message-passing layers) and Eq. (Init) becomes:

$$\mathbf{h}_V^0(i) = \text{INIT}_V\left(\mathbf{x}_V(i), \mathbf{p}_V(i)\right), \quad \mathbf{h}_E^0(i) = \text{INIT}_E\left(\mathbf{x}_E(i, j), \mathbf{p}_E(i, j)\right), \tag{15}$$

or alternatively, it can be performed at all layers, where Eqs. (Msg) and (Upd) become:

$$\mathbf{m}_V^t(i) = \bigoplus_{j \in \mathcal{N}(i)} \text{MSG}_V^t\left(\mathbf{h}_V^{t-1}(i), \mathbf{h}_V^{t-1}(j), \mathbf{h}_E^{t-1}(i, j), \mathbf{p}_V(i), \mathbf{p}_V(j), \mathbf{p}_E(i, j)\right) \tag{16}$$

$$\mathbf{h}_V^t(i) = \text{UPD}_V^t\left(\mathbf{h}_V^{t-1}(i), \mathbf{m}_V^t(i), \mathbf{p}_V(i)\right). \tag{17}$$

**Challenge 3 - Incorporating positional encodings into scale equivariant layers.** Although the above formulation allows for the preservation of valid permutation symmetries, special treatment is needed to preserve valid scaling symmetries as well. Since the positional encodings are fixed across datapoints, they do not have the same scaling symmetries as vertex/edge representations. *Therefore, message-passing components do not need to be equivariant to scaling of the positional encodings.* In particular, the following should hold: $\text{INIT}_V(q\mathbf{x}, \mathbf{p}) = q\text{INIT}_V(\mathbf{x}, \mathbf{p})$, $\text{INIT}_E(q\mathbf{e}, \mathbf{p}) = q\text{INIT}_E(\mathbf{e}, \mathbf{p})$, $\text{MSG}_V\left(q_x\mathbf{x}, q_y\mathbf{y}, q_x q_y^{-1}\mathbf{e}, \mathbf{p}_x, \mathbf{p}_y, \mathbf{p}_e\right) = q_x\text{MSG}_V(\mathbf{x}, \mathbf{y}, \mathbf{e}, \mathbf{p}_x, \mathbf{p}_y, \mathbf{p}_e)$ and $\text{UPD}_V(q_x\mathbf{x}, q_x\mathbf{m}, \mathbf{p}) = q_x\text{UPD}_V(\mathbf{x}, \mathbf{m}, \mathbf{p})$.

Therefore, Eqs. (15), (16), (17) introduce a new challenge (apart from those discussed in Section 5): being scale equivariant w.r.t. one argument and **not equivariant**. w.r.t. another one. Interestingly, our ScaleNet formalism can straightforwardly deal with this case, by defining the *canonicalisation* function for the non-symmetric argument to be the *identity* (since in the absence of symmetry equivalent points must be equal). To discern this from the classical scale invariant/equivariant layers, we will refer to it *augmented scale invariant/equivariant layer* and it is formalised as follows:

$$\text{AugScaleInv}^k(\mathbf{X}, \mathbf{p}) = \rho^k(\tilde{\mathbf{x}}_1, \ldots, \tilde{\mathbf{x}}_n, \mathbf{p}), \quad \tilde{\mathbf{x}}_i = \text{canon}(\mathbf{x}_i) \text{ or } \tilde{\mathbf{x}}_i = \text{symm}(\mathbf{x}_i) \quad (18)$$

$$\text{AugScaleEq} = \mathsf{f}^K \circ \cdots \circ \mathsf{f}^1, \quad \mathsf{f}^k(\mathbf{X}) = \left(\boldsymbol{\Gamma}_1^k\mathbf{x}_1, \ldots, \boldsymbol{\Gamma}_n^k\mathbf{x}_n\right) \odot \text{AugScaleInv}^k(\mathbf{X}, \mathbf{p}). \quad (19)$$

Concluding, we obtain the new learnable functions, by simply replacing scale equivariant with augmented scale equivariant layers:

$$\text{INIT}_V(\mathbf{x}, \mathbf{p}) = \text{AugScaleEq}(\mathbf{x}, \mathbf{p}), \quad \text{INIT}_E(\mathbf{e}, \mathbf{p}) = \text{AugScaleEq}(\mathbf{e}, \mathbf{p}), \quad (20)$$

$$\text{MSG}_V(\mathbf{x}, \mathbf{y}, \mathbf{e}, \mathbf{p}_x, \mathbf{p}_y, \mathbf{p}_e) = \text{AugScaleEq}([\mathbf{x}, \text{ReScaleEq}(\mathbf{y}, \mathbf{e})], [\mathbf{p}_x, \mathbf{p}_y, \mathbf{p}_e]), \quad (21)$$

$$\text{UPD}_V(\mathbf{x}, \mathbf{m}, \mathbf{p}_x) = \text{AugScaleEq}([\mathbf{x}, \mathbf{m}], \mathbf{p}_x). \quad (22)$$

### A.1.3 Edge Updates.

To respect the symmetries given by Eq. (5), for edges $(i, j)$, with layer$(i) \in \{2, \ldots, L-1\}$ the edge update function should have the following property: $\text{UPD}_E(q_x\mathbf{x}, q_y\mathbf{y}, q_x q_y^{-1}\mathbf{e}) = q_x q_y^{-1}\text{UPD}_E(\mathbf{x}, \mathbf{y}, \mathbf{e})$. Observe, that this is once again reminiscent of the rescaling problems discussed in Section 5, but with the difference that $q_y$ *appears both as a multiplier and as a divisor*. The most straightforward solution would be to elementwise invert $\mathbf{y}$ and then proceed as in the message function of Eq. (8), i.e. :

$$\text{UPD}_E(\mathbf{x}, \mathbf{y}, \mathbf{e}) = \text{ScaleEq}([\mathbf{e}, \text{ReScaleEq}(\mathbf{x}, \mathbf{1} \oslash \mathbf{y})]). \quad (23)$$

In our experiments though, we opt for a simpler solution due to the numerical instabilities (high-magnitude gradients) produced by the element-wise division and choose an invariant layer for the arguments $\mathbf{x}, \mathbf{y}$. Formally:

$$\text{UPD}_E(\mathbf{x}, \mathbf{y}, \mathbf{e}) = \text{AugScaleEq}(\mathbf{e}, \text{ScaleInv}(\mathbf{x}, \mathbf{y})). \quad (24)$$

When we include positional encodings into the edge updates, Eq. (Upd) (RHS) becomes:

$$\mathbf{h}_E^t(i, j) = \text{UPD}_E^t\left(\mathbf{h}_V^{t-1}(i), \mathbf{h}_V^{t-1}(j), \mathbf{h}_E^{t-1}(i, j), \mathbf{p}_V(i), \mathbf{p}_V(j), \mathbf{p}_E(i, j)\right), \quad (25)$$

and Eq. (24) is further modified into:

$$\text{UPD}_E(\mathbf{x}, \mathbf{y}, \mathbf{e}, \mathbf{p}_x, \mathbf{p}_y, \mathbf{p}_e) = \text{AugScaleEq}(\mathbf{e}, [\text{ScaleInv}(\mathbf{x}, \mathbf{y}), \mathbf{p}_x, \mathbf{p}_y, \mathbf{p}_e]). \quad (26)$$

### A.1.4 Updates on input and output vertices

So far, we have only discussed how vertex/edge representations corresponding to hidden neurons are updated, since Eqs. (4), (5) do not hold for input and output neurons. In particular, there exist no valid scaling or permutation symmetries for these neurons, while for the corresponding edges, permutations and scalings are unilateral - see Eq. (3). Therefore, we require the following:

$$\mathbf{h}_V'(i) = \mathbf{h}_V(i), \quad \ell = \text{layer}(i) \in \{0, L\}, \quad \text{(No equivariance)} \quad (27)$$

$$\mathbf{h}_E'(i, j) = q_\ell(\pi_\ell(i))\mathbf{h}_E(\pi_\ell(i), j), \quad \ell = \text{layer}(i) = 1, \quad (28)$$

$$\mathbf{h}_E'(i, j) = \mathbf{h}_E(i, \pi_{\ell-1}(j))q_{\ell-1}(\pi_{\ell-1}(j)), \quad \ell = \text{layer}(i) = L. \quad (29)$$

*This implies that the conditions for the learnable functions should change for i/o neurons.* In particular, the following should be **non-scale-equivariant**: $\text{INIT}_{V,0}$, $\text{MSG}_{V,0}$[7], $\text{UPD}_{V,0}$, $\text{INIT}_{V,L}$, $\text{UPD}_{V,L}$. The same should hold for $\text{MSG}_{V,L}$, but with the additional requirement of invariance to rescaling: $\text{MSG}_{V,L}(\mathbf{x}, q_y\mathbf{y}, q_y^{-1}\mathbf{e}, \mathbf{p}_x, \mathbf{p}_y, \mathbf{p}_e) = \text{MSG}_{V,L}(\mathbf{x}, \mathbf{y}, \mathbf{e}, \mathbf{p}_x, \mathbf{p}_y, \mathbf{p}_e)$

In traditional GNNs, where only permutation symmetries are taken into account, symmetry-breaking can be efficiently handled by positional encodings as discussed in Appendix A.1.2. However, this is not the case here, since positional encodings only break permutation symmetries, but not scaling symmetries. Instead, we resort to a simpler solution (albeit incurring an increased number of parameters): **we use different initialisation, message and update functions for input/output neurons.** $\text{INIT}_{V,0}$, $\text{MSG}_{V,0}$, $\text{UPD}_{V,0}$, $\text{INIT}_{V,L}$, $\text{UPD}_{V,L}$ are general function approximators (MLPs), while $\text{MSG}_{V,L}(\mathbf{x}, \mathbf{y}, \mathbf{e}, \mathbf{p}_x, \mathbf{p}_y, \mathbf{p}_e) = \text{MLP}(\mathbf{x}, \text{ReScaleEq}(\mathbf{y}, \mathbf{e}), \mathbf{p}_x, \mathbf{p}_y, \mathbf{p}_e)$.

As for the i/o edge representations, the following should hold at initialisation: $\text{INIT}_{E,1}(q_x\mathbf{e}, \mathbf{p}) = q_x\text{INIT}_{E,1}(\mathbf{e}, \mathbf{p})$, $\text{INIT}_{E,L}(q_y^{-1}\mathbf{e}, \mathbf{p}) = q_y^{-1}\text{INIT}_{E,L}(\mathbf{e}, \mathbf{p})$, and for the edge updates: $\text{UPD}_{E,1}(q_x\mathbf{x}, \mathbf{y}, q_x\mathbf{e}, \mathbf{p}_x, \mathbf{p}_y, \mathbf{p}_e) = q_x\text{UPD}_{E,1}(\mathbf{x}, \mathbf{y}, \mathbf{e}, \mathbf{p}_x, \mathbf{p}_y, \mathbf{p}_e)$, $\text{UPD}_{E,L}(\mathbf{x}, q_y\mathbf{y}, q_y^{-1}\mathbf{e}, \mathbf{p}_x, \mathbf{p}_y, \mathbf{p}_e) = q_y^{-1}\text{UPD}_{E,L}(\mathbf{x}, \mathbf{y}, \mathbf{e}, \mathbf{p}_x, \mathbf{p}_y, \mathbf{p}_e)$. Therefore, edge initialisation should be equivariant to the edge scale and therefore can remain as is - Eq. (20). The same holds for the edge updates - Eq. (26), which allows us to retain the same architecture in our experiments, although the interested reader may observe that one can use more powerful architectures since there is no need for rescaling here.

### A.1.5   Readout

Regarding the readout, we devise a permutation and scale invariant aggregator as follows:

$$\text{READ}_V(\mathbf{X}) = \text{MLP}\left([\text{DeepSets}(\tilde{\mathbf{x}}_1, \ldots, \tilde{\mathbf{x}}_n), \mathbf{X}_0, \mathbf{X}_L]\right), \tilde{\mathbf{x}}_i = \text{canon}(\mathbf{x}_i) \text{ or } \tilde{\mathbf{x}}_i = \text{symm}(\mathbf{x}_i). \tag{30}$$

This means that we aggregate all *scale canonicalised/symmetrised* representations of hidden neurons with a permutation invariant aggregator (*Deepsets* [81]), concatenate the result with all i/o neuron representations (that cannot be permuted or scaled) and transform the result with a universal function approximator. A simpler alternative is to only concatenate the output neuron representations (or also the input for the bidirectional version) since after sufficient message-passing iterations they are expected to have collected information from the entire graph. We include and evaluate both choices in our hyperparameter grid.

### A.2   Bidirectional ScaleGMN

Undoubtedly, using only forward neighbourhoods, although convenient and aligned with the computational graph of the input datapoint, may restrict the expressive power of ScaleGMN. This is self-explanatory since vertices receive information only from parts of the graph that belong to earlier layers. This might be detrimental, especially for equivariant tasks (operators). Additionally, it is contrary to the first works on weight space networks [54, 86], where vertices/edges collect information from the entire graph, while backward neighbourhoods are crucial for GMNs to express NP-NFN [86] as shown in Proposition 3 in Lim et al. [44].

However, making *bidirectional message passing* scale equivariant is challenging since backward messages have different symmetries than forward ones. In particular, preserving the same assumptions for the symmetries of vertex/edge representations, the input triplet to a message function $\mathbf{h}_V(i), \mathbf{h}_V(j), \mathbf{h}_E(j, i)$, with $j \in \mathcal{N}_B(i)$ and $\ell = \text{layer}(i)$ is equivalent to $q_\ell(i)\mathbf{h}_V(i), q_{\ell+1}(j)\mathbf{h}_V(j), q_{\ell+1}(j)\mathbf{h}_E(j, i)q_\ell^{-1}(i)$ (we have omitted permutations here for simplicity). Therefore, doing the same rescaling as before is inappropriate since it does not preserve the desired scaling coefficient $q_\ell(i)$.

There are several options one may follow to circumvent this discrepancy. However, most involve an elementwise division of a certain representation, which we experimentally observed to face numerical

---

[7]This is redundant in the forward ScaleGMN since input neurons do not have any forward neighbours. In the backward case - see Appendix A.2, we use a layer similar to $\text{MSG}_{V,L}$.

instabilities. The simplest one is to *add backward edges and initialise them with inverted features*:

$$\text{INIT}_E\left(\mathbf{x}_E\left(j,i\right)\right) = \mathbf{\Gamma}_E \mathbf{1} \oslash \mathbf{x}_E(j,i), \quad \mathbf{x}_E(j,i) = \tilde{\mathbf{W}}\left(j,i\right), \text{ if } 0 \leq \text{layer}\left(j\right) < \text{layer}\left(i\right) \leq L. \tag{31}$$

In this case, we have that $\text{INIT}_E\left(q_{\ell+1}(j)q_\ell^{-1}(i)\mathbf{x}_E\left(j,i\right)\right) = q_\ell(i)q_{\ell+1}^{-1}(j)\text{INIT}_E\left(\mathbf{x}_E\left(j,i\right)\right)$, and thus $(j,i)$ are initialised with the same symmetries as $(i,j)$. Now one can proceed as before with the message and update functions. In particular, we introduce a backward message function to discern the two message directions (alternatively this can be done with edge positional encodings to reduce the number of parameters):

$$\mathbf{m}_{V,\text{BW}}^t(i) = \bigoplus_{j \in \mathcal{N}_B(i)} \text{MSG}_{V,\text{BW}}^t(\mathbf{h}_V^{t-1}(i), \mathbf{h}_V^{t-1}(j), \mathbf{h}_E^{t-1}(j,i), \mathbf{p}_V(i), \mathbf{p}_V(j), \mathbf{p}_E(j,i))$$

$$\text{MSG}_{V,\text{BW}}(\mathbf{x}, \mathbf{y}, \mathbf{e}, \mathbf{p}_x, \mathbf{p}_y, \mathbf{p}_e) = \text{AugScaleEq}\left([\mathbf{x}, \text{ReScaleEq}\left(\mathbf{y}, \mathbf{e}\right)], [\mathbf{p}_x, \mathbf{p}_y, \mathbf{p}_e]\right).$$

It is easy to see that:

$$\text{MSG}_{V,\text{BW}}(q_x\mathbf{x}, q_y\mathbf{y}, q_y^{-1}q_x\mathbf{e}, \mathbf{p}_x, \mathbf{p}_y, \mathbf{p}_e) = q_x\text{MSG}_{V,\text{BW}}(\mathbf{x}, \mathbf{y}, \mathbf{e}, \mathbf{p}_x, \mathbf{p}_y, \mathbf{p}_e).$$

Now to incorporate backward messages into the update function we simply concatenate the outputs of the two message functions and apply a scale equivariant layer as usual:

$$\mathbf{h}_V^t(i) = \text{UPD}_V^t\left(\mathbf{h}_V^{t-1}(i), \mathbf{m}_{V,\text{FW}}^t(i), \mathbf{m}_{V,\text{BW}}^t(i), \mathbf{p}_V(i)\right),$$

$$\text{UPD}_V\left(\mathbf{x}, \mathbf{m}_{\text{FW}}, \mathbf{m}_{\text{BW}}, \mathbf{p}_x\right) = \text{AugScaleEq}\left([\mathbf{x}, \mathbf{m}_{\text{FW}}, \mathbf{m}_{\text{BW}}], \mathbf{p}_x\right). \tag{32}$$

Once, again the desired symmetries are preserved:

$$\text{UPD}_V(q_x\mathbf{x}, q_x\mathbf{m}_{\text{FW}}, q_x\mathbf{m}_{\text{BW}}, \mathbf{p}_x) = q_x\text{UPD}_V(\mathbf{x}, \mathbf{m}_{\text{FW}}, \mathbf{m}_{\text{BW}}, \mathbf{p}_x).$$

**Bidirectional ScaleGMN for sign symmetries.** Importantly, the above edge weight inversion of Eq. (31) is not necessary when dealing with sign symmetries. In particular, for any $q \in \{-1, 1\}$ it holds that $\frac{1}{q} = q$. Therefore, one can proceed with the above formulation but initialise the edge weights as before -Eq. (12). We found this crucial in our experimental evaluation since weight inversion led to an unusual distribution of input features: since typically the weight distribution was nearly Gaussian, the distribution of the inverses was close to *reciprocal normal distribution*, a bimodal distribution with undefined mean and higher-order moments. This frequently led to numerical instabilities as well (albeit less so compared to computing reciprocals of neuron/edge representations), which explains why it was easier to train bidirectional models for sign symmetric NNs compared to positive-scale symmetric ones.

### A.3   Extension to CNNs

A similar methodology to Section 5 can be applied to Convolutional Neural Networks with only a few changes. As observed by prior works on metanetworks (Zhou et al. [85], Kofinas et al. [33]), the permutation symmetries in a CNN arise similarly to MLPs; permuting the filters (output channels) of a hidden convolutional layer, while also permuting the input channels of each filter in the subsequent layer does not alter the underlying function of the network. This is easy to see since convolution is nothing else but a linear layer with weight sharing. Weight sharing constrains the valid permutations as far as pixels/input coordinates are concerned (e.g. one cannot permute the representations corresponding to different pixels), but allows for permutations of input/output channels, except for the input/output layers as always.

Similar rules apply to scaling: one may scale the output channels of a hidden convolutional layer, while also scaling the input channels of each filter in the subsequent layer as per Proposition 4.1. Note here that, again, all weights corresponding to the same input channel can only be scaled with the same multiplier for all pixels/input coordinates, due to weight sharing. Below, we describe our implementation for all CNN layers, which closely follows [33].

**Convolution.** Being the main building block of CNNs, convolutional layers consist of kernels and biases. For a hidden layer $i$ we can write the former as $\mathbf{W}_i \in \mathbb{R}^{d_\ell \times d_{\ell-1} \times k_\ell}$ and the latter as $\mathbf{b}_i \in \mathbb{R}^{d_\ell}$ following the FFNN notation. Aligning with the procedure of [33], we construct one vertex for

each i/o channel and the input graph has the following node and edge features: $\mathbf{x}_V(i) = \tilde{\mathbf{b}}(i) \in \mathbb{R}$, $\mathbf{x}_E(i,j) = \tilde{\mathbf{W}}(i,j) \in \mathbb{R}^{w_{\max} \cdot h_{\max}}$ respectively, where $w_{\max}, h_{\max}$ are the maximum width and height of the kernels across layers (zero-padding is used to allow for fixed-size edge representations across all vertices in the graph). Again here, the positional encodings are responsible for the *permutation symmetry breaking mechanisms* within the CNN. As with FFNNs, the CNN input and output neurons are not permutable, while the hidden neurons are permutable only within the layer they belong. Given that filter positions are not permutable and are associated with a single neuron, *they are represented as vectors on the edge features*.

**Average Pooling.** Average pooling is commonly placed after the last convolutional layer (typical in modern CNNs instead of flattening + linear layer to allow for variable-size images) and is responsible for pooling the output feature map in a single feature vector. First off, since it is a linear operation, it is amenable to scaling symmetries. It is easy to see this by perceiving average pooling as a $KxK$ convolution with no learnable parameters (equal to $\frac{1}{K^2}$, where $K$ the dimension of the image, and a single output channel. Hence, no additional modifications are required, similarly to [33].

**Linear layer.** Commonly used as the last layer of CNNs to extract a vector representation of an image, linear layers require a different approach within CNNs compared to the one in MLPs, as in the former case the edge features are vectors, while in the latter scalars. Aligning with the method proposed by [33], the simplest solution is to zero-pad the linear layers to the maximum kernel size and consequently flatten them to meet the dimension of the rest of the edge features. By handling linear layers this way, no additional measures are needed to ensure permutation and scale equivariance.

### A.4  Implementation Details

#### A.4.1  INR Classification

**Datasets:** We evaluated our method on three INR datasets. Provided as open-source by Navon et al. [54], the datasets MNIST and FashionMNIST contain a single INR for each image of the datasets MNIST [38] and FashionMNIST [76] respectively. The selection of these datasets was encouraged by the fact that they were also selected by prior works, establishing them as a useful first benchmark on INR metanetworks. As a third INR dataset, we use CIFAR-10, publicly available by Zhou et al. [85], which contains one INR per image from CIFAR10 [35]. Regarding the last dataset, the authors also provide an augmented training dataset, consisting of 20 additional copies of each INR network, but with different initializations. Training on such a bigger dataset allows them to achieve better results, probably because it allows them to counteract overfitting. To demonstrate the capabilities of our method we train ScaleGMN both on the "original" dataset as well as on the augmented one, achieving better results in both cases. In the latter case we carefully follow the training and evaluation procedure used in [85], i.e. we train for 20000 steps and evaluate every 3000 steps. In all the above datasets, we use the same splits as in prior works, we train for 150 epochs and report the test accuracy based on the validation split, following the same procedure as Navon et al. [54].

**Hyperparameters:** The GNN used for ScaleGMN is a traditional Message Passing Neural Network [24], with carefully designed update and message functions that ensure sign equivariance. We optimise the following hyperparameters: batch size in $\{64, 128, 256\}$, hidden dimension for node/edge features in $\{64, 128, 256\}$. We also search learning rates in $\{0.001, 0.0005, 0.0001\}$, weight decay in $\{0.01, 0.001\}$, dropout in $\{0, 0.1, 0.2\}$ and number of GNN layers in $\{2, 3, 4, 5\}$. Moreover, we experiment with using only vertex positional encodings or also employing edge positional encodings. We apply layer normalization within each MLP and use skip connections between each GNN layer. The last two steps were proven valuable to stabilise training. Finally, for each MLP within the architecture we use SiLU activation function, one hidden layer and no activation function after the last layer. All the experiments were conducted on NVIDIA GeForce RTX 4090.

As mentioned above, the core of all building blocks, i.e. Scale Invariant Net is designed to be invariant to sign flips. In our experiments, we explore two means of designing sign equivariance, either by elementwise absolute value $|\mathbf{x}|$ (canonicalization only in 1-dimension), or symmetrisation, $\text{symm}(\mathbf{x}) = \text{MLP}(\mathbf{x}) + \text{MLP}(-\mathbf{x})$. Although the former leads to reduced expressivity, in some cases we observed better performance. A possible explanation could point to the extra training parameters that are added because of the additional MLP.

**Results:** As shown in table Table 1, our method is able to push the state-of-the-art in all the datasets considered. Notably, many additional techniques that were employed by previous works in order to

achieve good results were not required in our method. Specifically, probe features when inserted into the node features as in [33] were proven to boost the performance much higher, and could potentially enhance the performance of ScaleGMN too. In the reported results we compare with variants of [33] that do not use probe features and rely solely on the parameters of the input network, although our method outperforms even the probe features variants. Moreover, [85] and [33] also apply random Fourier features during the initialization of the weight and bias features, which in our experiments showcased better performance when used in a traditional MPNN framework, but were not employed for our method. In our framework, we also do not include any normalization of the input as conducted by previous works, where the normalization is applied either on parameter level ([85], [54]) or on layer level [33], computing the mean and standard deviation. Although this step typically leads to better performance, it could not be applied in our scale and permutation equivariant framework. Nevertheless, even without all the above techniques, ScaleGMN achieves remarkable results. Finally, opting for a Transformer, as in [33], instead of a GNN layer is also an orthogonal to our work addition that could enhance the performance of ScaleGMN.

### A.4.2 Predicting CNN Generalization from weights

For evaluating our method on the CNN Generalization task, we select CIFAR-10-GS and SVHN-GS datasets from [74], following the evaluation of prior works (Zhou et al. [85], Zhou et al. [86], Lim et al. [44], Kofinas et al. [33]). The above datasets contain CNNs trained with ReLU and Tanh activation functions. In order to assess our method separately for each of these activation functions, as they introduce different symmetries, we distinguish two different types of experiments. In the first case we deviate from the path of previous evaluations and split each of these datasets into two subsets. The first one contains CNNs trained with ReLU activation function and the second networks with Tanh. In order to compare with previous methods, we train them on each subset and report Kendall's correlation $\tau$ metric [30] as originally proposed by Zhou et al. [85].

**Heterogeneous activation functions:** In the second case, we evaluate ScaleGMN on the whole CIFAR-10-GS and SVHN-GS datasets. We are able to do so, as our method does not impose any limitations regarding the homogeneity of the activation functions of the input neural networks. The sole necessary modification lies in the construction of the invariant layer. Specifically, following the notation of Eq. (Scale Equiv. Net), each equivariant layer $\mathsf{f}^k$ is now equipped with two invariant nets, $\mathsf{ScaleInv}_{relu}^k$ and $\mathsf{ScaleInv}_{tanh}^k$, to be applied on the datapoints with ReLU and Tanh activation functions respectively. The rest of the architecture remains the same.

**Hyperparameters:** For the ReLU datasets we implement scale equivariant update and message functions, while for the Tanh sign equivariant. In all of our experiments we use MSE training loss. Regarding the hyperparameters we search: batch size in $\{32, 64, 128\}$, hidden dimension for node/edge features in $\{64, 128, 256\}$. We also search learning rates in $\{0.001, 0.0005, 0.0001\}$, weight decay in $\{0.01, 0.001\}$, dropout in $\{0, 0.1, 0.2\}$ and number of GNN layers in $\{2, 3, 4, 5\}$. Again, we experiment with using only node positional encodings or also employing edge positional encodings and apply layer normalization within each MLP. For the Tanh datasets, we also experiment with the applying canonicalization or symmetrization of the sign symmetry. Finally, for the experiments with heterogeneous activation functions, we apply the same hyperparameter search as above.

### A.4.3 INR editing

To evaluate the performance of ScaleGMN on equivariant tasks we opted for the task of INR editing. Aligning with the setup of [33], we utilize the MNIST INR dataset provided by Navon et al. [54] and evaluate our method on dilating the encoded MNIST digits. Specifically, for each image $i$ of the dataset, where $i \in [N]$, we compute the ground truth dilated image using the OpenCV image processing library and denote it as $\hat{f}_i$, where $f_i : \mathbb{R}^2 \to \mathbb{R}$. Let $\boldsymbol{\theta}_i$ denote the INR parameters of the $i$-th image and $f_{\text{SIREN}}$ the encoded function. Then, $f_{\text{SIREN}}(x, y; \boldsymbol{\theta}_i)$ is the output of the INR at the $(x, y)$ coordinates, when parameterized by $\boldsymbol{\theta}_i$. The updated INR weights are computed as: $\boldsymbol{\theta}_i' = \boldsymbol{\theta}_i + \gamma \cdot \text{ScaleGMN}(\boldsymbol{\theta}_i)$, where $\gamma$ a learned scalar initialized to 0.01. Finally, we minimize the mean squared error on the function space, that is:

$$\mathcal{L} = \frac{1}{N \cdot d^2} \sum_{i=1}^{N} \sum_{x,y}^{d} \|f_{\text{SIREN}}(x, y; \boldsymbol{\theta}_i') - f_i(x, y)\|_2^2 \tag{33}$$

Simply put, we compare the reconstructed new INR with the dilated ground truth image. Again, we do not apply any normalizations nor feed the model samples of the encoded input image (probe features).

**Hyperparameters:** The GNN used for ScaleGMN is a traditional Message Passing Neural Network [24], with carefully designed update and message functions that ensure sign equivariance. In contrast to the above experiments we do not add a final readout layer, as we do not compute any final graph embedding. We optimise the following hyperparameters: batch size in $\{64, 128, 256\}$, hidden dimension for node/edge features in $64, 128, 256$ and number of GNN layers in $\{8, 9, 10, 11\}$. We also search learning rates in $\{0.001, 0.0005, 0.0001\}$, weight decay in $\{0.01, 0.001\}$ and dropout in $\{0, 0.1, 0.2\}$. Moreover, we experiment with using only vertex positional encodings or also employing edge positional encodings. We apply layer normalization within each MLP and use skip connections between each GNN layer. The last two steps were proven valuable to stabilise training. Finally, for each MLP within the architecture we use SiLU activation function, one hidden layer and no activation function after the last layer. The learned scalar $\gamma$, also used in previous work Zhou et al. [85], was always initialized at the same value $0.001$ and consistently led to better results. All the experiments were conducted on NVIDIA GeForce RTX 4090.

**Results:** As shown in Table 3, the bidirectional variant achieves better results than all the previous works on metanetworks, without incorporating any of the additional techniques described above. Notably, ScaleGMN-B surpasses the performance of the graph-based permutation equivariant baseline NG-GNN equipped with 64 probe features [33]. Interestingly, our method demonstrated much better results as we added more GNN layers. This experimental observation possibly corroborates with the findings of Theorem 5.2. Regarding our forward variant, we can see that although it is able to perform on par with the previous baselines, it struggles to match the results achieved by the bidirectional variant. This behaviour, observed exclusively in this task, is theoretically anticipated for equivariant tasks. Specifically, in the forward case, neurons of layer $\ell$ only receive information from layers $\ell' < \ell$, which hinders the computation of meaningful embeddings for nodes of the first layers. Taking into account that the INR models are rather shallow (only 2 hidden layers of 32 neurons each), then a big proportion of the whole set of nodes is being updated with less meaningfull information.

### A.4.4 Extra symmetries of the sine activation function

Regarding the `sine` activation function, one can easily show the presence of an additional symmetry due to the periodicity of harmonic functions. To see this, recall from Eq. (1) the formula for a feedforward $L$-layer SIREN:

$$\mathbf{x}_0 = \mathbf{x}, \quad \mathbf{x}_\ell(i) = \sin\left(\omega_0 \mathbf{W}_\ell(i,:) \mathbf{x}_{\ell-1} + \mathbf{b}_\ell(i)\right).$$

We will prove that the following guarantees function preservation:

$$\mathbf{W}'_\ell = \mathbf{Q}_\ell \mathbf{W}_\ell, \ \mathbf{b}'_\ell = \mathbf{Q}_\ell \mathbf{b}_\ell + \mathbf{O}_\ell \pi \ \Rightarrow (\mathbf{W}'_\ell, \mathbf{b}'_\ell)^L_{\ell=1} \simeq (\mathbf{W}_\ell, \mathbf{b}_\ell)^L_{\ell=1}, \tag{34}$$

with $\mathbf{Q}_\ell$ defined as diagonal sign matrices, $\mathbf{O}_\ell = 2\mathbf{M}_\ell + \text{inv-sign}(\mathbf{Q}_\ell)$, where $\mathbf{M}_\ell = \text{diag}\left(m_\ell(1), \ldots m_\ell(d_\ell)\right)$, $m_\ell(i) \in \mathbf{Z}$ and we defined $\text{inv-sign}(q) = \begin{cases} 0, \ q \geq 0 \\ 1, \ q < 0. \end{cases}$ for compactness (we have omitted permutation matrices here for brevity). Observe here that the matrix transformations are "single-sided", contrary to Eq. (3). We show by induction that, when Eq. (34) holds, then $\mathbf{x}'_\ell = \mathbf{x}_\ell$.

*Base case:* $\mathbf{x}'_0 = \mathbf{x}_0$ by defintion. *Induction step:* Assume $\mathbf{x}'_{\ell-1} = \mathbf{x}_{\ell-1}$. In order to have $\mathbf{x}'_\ell = \mathbf{x}_\ell, \ \forall \mathbf{x} \in \mathcal{X}$, it suffices that $\exists m_\ell(i) \in \mathbb{Z}, q_\ell(i) \in \{-1, 1\}$:

$$\omega_0 \mathbf{W}'_\ell(i,:) \mathbf{x}_{\ell-1} + \mathbf{b}'_\ell(i) = \left(\omega_0 \mathbf{W}_\ell(i,:) \mathbf{x}_{\ell-1} + \mathbf{b}_\ell(i)\right) q_\ell(i) + \left(2m_\ell + \text{inv-sign}(q_\ell(i))\right)\pi$$

$$\Leftarrow \mathbf{W}'_\ell(i,:) = q_\ell(i) \mathbf{W}_\ell(i,:), \quad \mathbf{b}'_\ell(i) = q_\ell(i) \mathbf{b}_1(i) + \left(2m_\ell(i) + \text{inv-sign}(q_\ell(i))\right)\pi.$$

This can be straightforwardly extended to include permutations.

To account for those symmetries, in this paper, we opted for a simple solution, based on the observation *that they cannot exist if the biases are bounded*. In particular, *if* $\mathbf{b}'_\ell(i), \mathbf{b}_\ell(i) \in (-\pi/2, \pi/2]$, *then* $m$ *must be 0 and q must be 1 in all cases*: For $q_\ell(i) = 1$, since $0 \leq |\mathbf{b}_\ell(i) - \mathbf{b}_\ell(i)| < \pi$, it must be that $0 \leq |2m_\ell(i)| < 1 \Leftrightarrow 0 \leq |m_\ell(i)| < 1/2 \Rightarrow m = 0$. For $q_\ell(i) = -1$, since $-\pi < \mathbf{b}_\ell(i) + \mathbf{b}_\ell(i) \leq \pi$, it must be that $-1 < 2m_\ell(i) + 1 \leq 1 \Leftrightarrow -1 < m_\ell(i) \leq 0 \Rightarrow m_\ell(i) = 0$. Further, since the

---

**Algorithm 1:** Bias shift (elimination of symmetries due to periodicity)

---

1: **Input:** $b$: bias $\mathbf{b}_\ell(i)$, $\mathbf{w}$: weight row $\mathbf{W}_\ell(i,:)$
2: **if** $b < 0$ **then**
3:   $b_1 \leftarrow -b; \mathbf{w}_1 \leftarrow -\mathbf{w}; q_1 \leftarrow -1$ // $\sin(\omega_0 \mathbf{w}_1^\top \mathbf{x}_{\ell-1} + b_1) = -\sin(\omega_0 \mathbf{w}^\top \mathbf{x}_{\ell-1} + b)$
4: **else**
5:   $b_1 \leftarrow b; \mathbf{w}_1 \leftarrow \mathbf{w}; q_1 \leftarrow 1$
6: **end if**// Now $b_1 \geq 0$.
7: **if** $b_1 > 2\pi$ **then**
8:   $b_2 \leftarrow b_1 \mod 2\pi; \mathbf{w}_2 \leftarrow \mathbf{w}_1; q_2 \leftarrow 1$
     // $b_1 = 2\pi k + b_2 \Rightarrow \sin(\omega_0 \mathbf{w}_2^\top \mathbf{x}_{\ell-1} + b_2) = \sin(\omega_0 \mathbf{w}_1^\top \mathbf{x}_{\ell-1} + b_1)$
9: **else**
10:   $b_2 \leftarrow b_1; \mathbf{w}_2 \leftarrow \mathbf{w}_1; q_2 \leftarrow 1$
11: **end if**// Now $0 \leq b_2 \leq 2\pi$.
12: **if** $\pi < b_2 \leq 2\pi$ **then**
13:   $b_3 \leftarrow b_2 - \pi; \mathbf{w}_3 \leftarrow \mathbf{w}_2; q_3 \leftarrow -1$ // $\sin(\omega_0 \mathbf{w}_3^\top \mathbf{x}_{\ell-1} + b_3) = -\sin(\omega_0 \mathbf{w}_2^\top \mathbf{x}_{\ell-1} + b_2)$
14: **else**
15:   $b_3 \leftarrow b_2; \mathbf{w}_3 \leftarrow \mathbf{w}_2; q_3 \leftarrow 1$
16: **end if**// Now $0 \leq b_3 \leq \pi$.
17: **if** $0 \leq b_3 \leq \pi/2$ **then**
18:   $b_4 \leftarrow b_3; \mathbf{w}_4 \leftarrow \mathbf{w}_3; q_4 \leftarrow 1$
19:   **Return:** $(b_4, \mathbf{w}_4, q_4)$
20: **else**
21:   $b_4 \leftarrow b_3 - \pi; \mathbf{w}_4 \leftarrow \mathbf{w}_3; q_4 \leftarrow -1$ // $\sin(\omega_0 \mathbf{w}_4^\top \mathbf{x}_{\ell-1} + b_4) = -\sin(\omega_0 \mathbf{w}_3^\top \mathbf{x}_{\ell-1} + b_3)$
22:   **Return:** $(b_4, \mathbf{w}_4, q_4)$
23: **end if**// Now $-\pi/2 < b_4 \leq \pi/2$.
24: **Output:** New bias $\tilde{\mathbf{b}}_\ell(i) \leftarrow \prod q_i * b_4$ and new weight row $\tilde{\mathbf{W}}_\ell(i,:) \leftarrow \prod q_i * w_4$.
     // $\sin(\prod q_i \omega_0 \mathbf{w}_4^\top \mathbf{x}_{\ell-1} + \prod q_i b_4) = \prod q_i q_4 \sin(\omega_0 \mathbf{w}_3^\top \mathbf{x}_{\ell-1} + b_3) = \cdots = \sin(\omega_0 \mathbf{w}^\top \mathbf{x}_{\ell-1} + b)$

---

equality holds only when both biases are equal to $\pi/2$, then the case $q_\ell(i) = -1, m = 0$ reduces to $q_\ell(i) = 1, m = 0$.

**Overall, by constraining the biases in the $(-\pi/2, \pi/2]$ interval, these symmetries are eliminated.** In all datasets considered, this constraint was already satisfied. For completeness, we provide Algorithm 1 that illustrates a method to shift biases in the desired interval.

## A.5   Additional ablation studies

### A.5.1   Distribution of Scaling coefficients

Empirically, accounting for a given symmetry might not prove fruitful if the given dataset is by construction canonicalised, e.g. if all parameters were already positive or negative in the presence of sign symmetries. To ensure the occurrence of the studied symmetries within the used datasets, we explore their statistics. Regarding the positive-scaling symmetry, the *distribution of norms of weights and biases within each network layer* in the CIFAR-10-GS-ReLU dataset is shown in Fig. 1. Concerning sign symmetries, we similarly plot the *distribution of the sign values* for the MNIST-INR dataset in Fig. 2 and the CIFAR-10-GS-tanh dataset in Fig. 3. Observe that in the former case, the distribution has relatively high variance, showcasing that the weights/biases are not by training scale-normalised. Even more prominently, in the latter case, the probability of each sign value is close to $1/2$ which corroborates our claims on the need for sign equivariance/invariance.

## A.6   Additional Considerations

In the following section, we explore key questions raised by the reviewers during the evaluation process. These questions touch upon interesting aspects of our methodology that merit in-depth exploration.

**Complexity and performance (runtime) degradation between ScaleGMN and ScaleGMN-B**
In the bidirectional case, we add backward edges to our graph and introduce a backward message

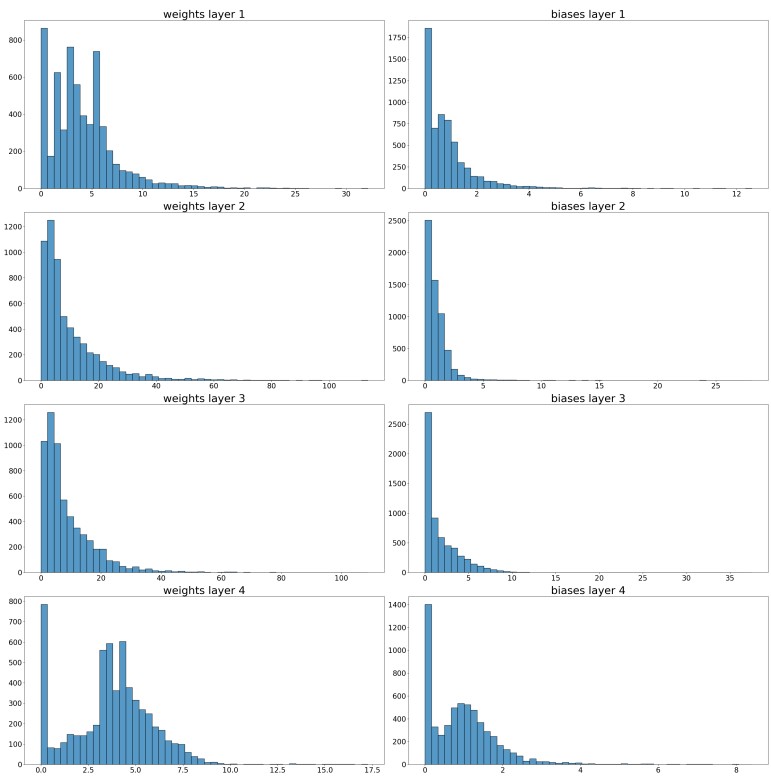

Figure 1: Norm distribution on the weights and biases in the CIFAR-10-GS-ReLU dataset.

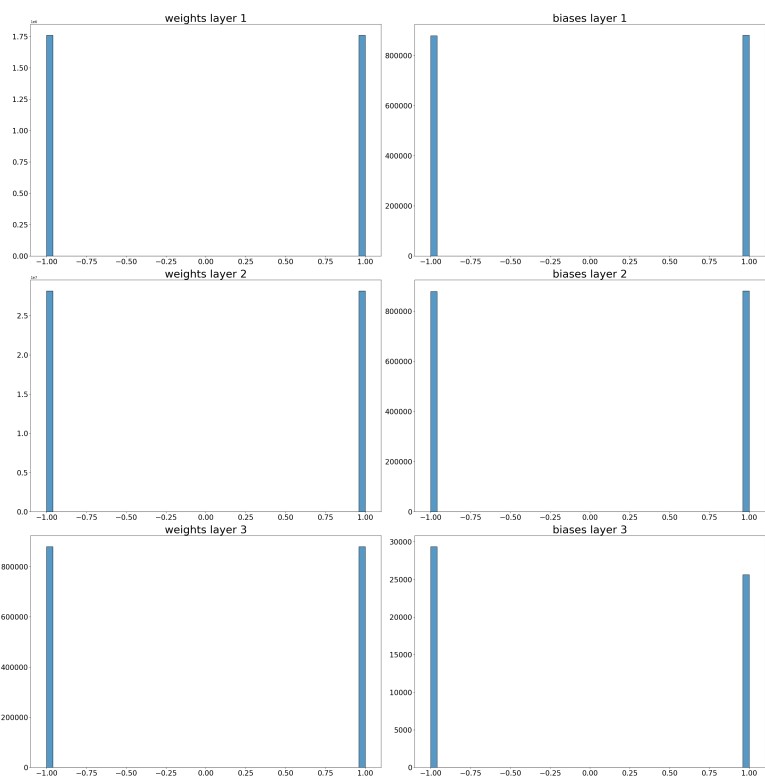

Figure 2: Sign distribution of the weights and biases in the MNIST INR dataset.

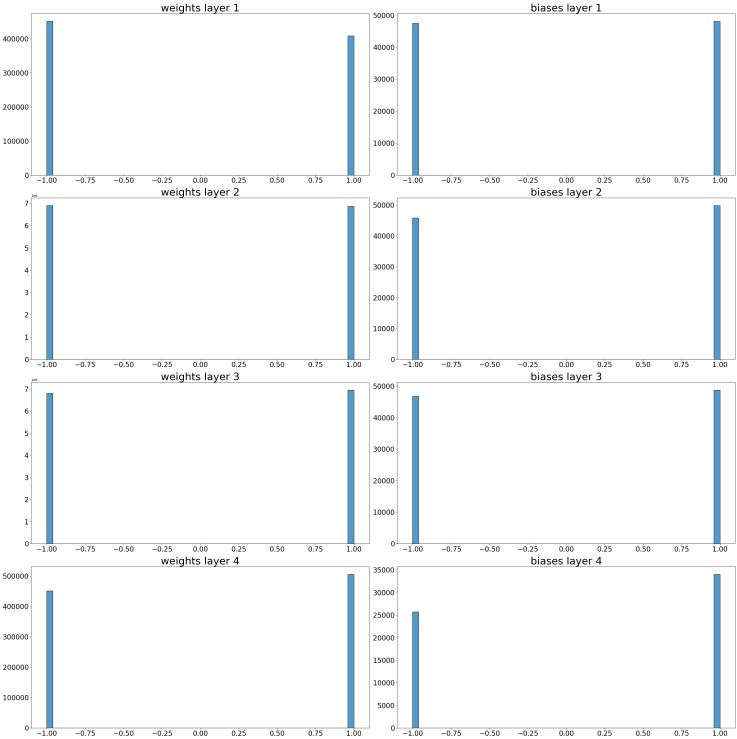

Figure 3: Sign distribution of the weights and biases in the CIFAR-10-GS-tanh dataset.

function to discern the two message directions. Subsequently, we concatenate the outputs of the two message functions and apply the UPD function. Consequently, the additional complexity of ScaleGMN-B is introduced solely by the extra message function, with a complexity of $O(E)$. Given that ScaleGMN has the complexity of a standard GNN model $O(V + E)$, the final complexity of ScaleGMN-B is $O(V + 2E)$.

**Trainable parameters: GMN vs ScaleGMN(-B)?** Going from a GMN model to ScaleGMN requires adapting the MSG and UPD functions to be scale equivariant, leading to more learnable parameters, as opposed to using plain MLPs. Another design choice of ScaleGMN that introduces more parameters is using different MLPs for the I/O nodes (A.1.4).

**Are there any limitations on the choice of activations of the ScaleGMN network?** Importantly, our method does *not* impose any limitations on the choice of the activation functions. We are able to select any activation function, because these are only applied within the MLPs of the invariant modules. As discussed in Section 5, the MLPs (equipped with non-linearities) are only applied after the canonicalization/symmetrization function. In case one chose to place activations in a different computational part of ScaleGMN, this would indeed limit their options so as not to compromise scale equivariance. However, this is not the case in our method.

**Canon/symm mappings for all activation functions.** In general, we cannot guarantee that canonicalization/symmetrization mappings are easy to construct for any activation function, since it is currently unknown if we can provide a general characterisation of the possible symmetries that may arise. Our results extend to all positively homogeneous activations, $\sigma(\lambda x) = \lambda\sigma(x), \lambda > 0$, and all odd ones, $\sigma(-x) = -\sigma(x)$. We refer the reviewer to Table 1 of Godfrey et al. [25], where we can see that LeakyReLU also falls in the first category. Regarding polynomial activations, which demonstrate non-zero scaling symmetries, one option would be: (1) norm division to canonicalise scale, and (2) sign symm/canon as discussed in this paper. The above, cover a fairly broad spectrum of common activation functions.

## A.7 Deferred Proofs

### A.7.1 Characterisation of symmetries

First we restate Proposition 4.1 and provided the detailed conditions for the activation function $\sigma$.

**Proposition A.1** (Proposition 4.1 restated). *Consider an activation function $\sigma : \mathbb{R} \to \mathbb{R}$.*

- *(Lemma 3.1. [25].) If the matrix $\sigma(\mathbf{I}_d)$ is invertible, then, for any $d \in \{1, 2, \dots\}$, there exists a (non-empty) group of matrices, called the **intertwiner group of** $\sigma_d$ defined as follows:*

$$I_{\sigma,d} = \{\mathbf{A} \in \mathbb{R}^{d \times d} : \text{invertible} \mid \exists\, \mathbf{B} \in \mathbb{R}^{d \times d} \text{ invertible, such that: } \sigma(\mathbf{A}\mathbf{x}) = \mathbf{B}\sigma(\mathbf{x})\},$$

  *and a group homomorphism $\phi_{\sigma,d} : I_{\sigma,d} \to GL_d(\mathbb{R})$, such that $\mathbf{B} = \phi_{\sigma,d}(\mathbf{A}) = \sigma(\mathbf{A})\sigma(\mathbf{I}_d)^{-1}$.*

- *(Theorem E.14. [25].) Additionally, if $\sigma$ is non-constant, non-linear and differentiable on a dense open set with finite complement, then every $\mathbf{A} \in I_{\sigma,d}$ is of the form $\mathbf{PQ}$, where $\mathbf{P}$: permutation matrix and $\mathbf{Q} = diag(q_1, \dots q_d)$ diagonal matrix, with $\phi_{\sigma,d}(\mathbf{A}) = \mathbf{P}diag(\phi_{\sigma,1}(q_1), \dots \phi_{\sigma,1}(q_d))$*

Note that $\mathbf{I}_d$ is the identity matrix $GL_d(\mathbb{R})$ is the general linear group. The above two statements are proven in [25]. The exact form of $\phi_{\sigma,d}(\mathbf{A})$ is not precisely given, but it is easy to derive: $\phi_{\sigma,d}$ is a group homomorphism and therefore $\phi_{\sigma,d}(\mathbf{PQ}) = \phi_{\sigma,d}(\mathbf{P})\phi_{\sigma,d}(\mathbf{Q}) = \mathbf{P}diag(\phi_{\sigma,1}(q_1), \dots \phi_{\sigma,1}(q_d))$, where we also used Lemma E.10. from [25] which states that $\phi_{\sigma,d}(\mathbf{P}) = \mathbf{P}$ for permutation matrices.

**Proposition A.2** (restated). *Hyperbolic tangent activation function $\sigma(x) = \tanh(x)$ and SIREN activation function, i.e. $\sigma(x) = \sin(\omega x)$, satisfy the conditions of Proposition 4.1, when (for the latter) $\omega \neq k\pi, k \in \mathbb{Z}$. Additionally, $I_{\sigma,d}$ contains **signed permutation matrices** of the form $\mathbf{PQ}$, with $\mathbf{Q} = diag(q_1, \dots, q_d)$, $q_i = \pm 1$ and $\phi_{\sigma,d}(\mathbf{PQ}) = \mathbf{PQ}$.*

*Proof.* **Example 1: $\sigma(x) = \sin(\omega x)$, with $\omega > 0$ constant**:

This is the most common activation function for INRs [70]. We can easily find the conditions under which $\sigma(\mathbf{I}_d)$ is invertible invoking Lemma E.7. from Godfrey et al. [25]. In particular, the latter states that if $\sigma(1) \neq \sigma(0)$, $\sigma(1) \neq -(d-1)\sigma(0)$, then $\sigma(\mathbf{I}_d)$ is invertible.

In our case, $\sigma(0) = \sin(\omega 0) = 0, \sigma(1) = \sin(\omega)$. Therefore, it should hold that $\omega \neq k\pi, k \in \mathbb{Z}$, otherwise $\sigma(\mathbf{I}_d)$ is an all-zeros matrix. Additionally, $\sigma$ is non-constant, non-linear and differentiable everywhere. Thus, as dictated by Proposition 4.1, we can characterise the intertwiner group, by identifying the function $\phi_{\sigma,1}(a)$. In particular, $b = \phi_{\sigma,1}(a)$ if the following holds:

$$\sin(\omega a x) = b\sin(\omega x) \Rightarrow \sin^2(\omega a x) = b^2 \sin^2(\omega x) \Rightarrow 1 - \cos^2(\omega a x) = b^2 \sin^2(\omega x)$$

$$\omega a \cos(\omega a x) = b\omega \cos(\omega x) \Rightarrow a^2 \cos^2(\omega a x) = b^2 \cos^2(\omega x),$$

where in the second line we took the derivative of each side since the equation holds for all $x \in \mathbb{R}$ and divided by $\omega \neq 0$. Now summing up the two equations we obtain:

$$1 + \cos^2(\omega a x)(a^2 - 1) = b^2 \Rightarrow \cos^2(\omega a x)(a^2 - 1) = b^2 - 1 \Rightarrow a^2 - 1 = b^2 - 1 = 0.$$

The above is true because if $a^2 \neq 1$, then one would conclude that $\cos^2(\omega a x) = $ constant. Thus, $a = \pm 1$ and $b = \pm 1$. Additionally $a$ cannot be different than $b$, since in that case we would have e.g. $\sin(\omega x) = -\sin(\omega x) \Rightarrow \sin(\omega x) = 0$. Therefore:

$$\phi_{\sigma,1}(a) = a, \quad a = \{-1, 1\} \text{ and } \phi_{\sigma,d}(\mathbf{PQ}) = \mathbf{PQ}, \quad \mathbf{PQ} : \text{signed permutation matrices.} \quad (35)$$

**Example 2: $\sigma(x) = \tanh(x)$:**

We can easily verify that $\sigma(\mathbf{I}_d)$ is invertible invoking the same Lemma E.7. from Godfrey et al. [25] as before. We have, $\sigma(0) = \tanh(0) = 0, \sigma(1) = \tanh(1) = \frac{e^2-1}{e^2+1} \neq 0$. Additionally, $\sigma$ is non-constant, non-linear and differentiable everywhere. Thus, again as dictated by Proposition 4.1, we

will characterise the intertwiner group, by identifying the function $\phi_{\sigma,1}(a)$. In particular, $b = \phi_{\sigma,1}(a)$ if the following holds:

$$\tanh(ax) = b\tanh(x) \Rightarrow \tanh^2(ax) = b^2\tanh^2(x)$$
$$a\big(1 - \tanh^2(ax)\big) = b\big(1 - \tanh^2(x)\big),$$

where in the second line we took the derivative of each side since the equation holds for all $x \in \mathbb{R}$. Now summing up the two equations we obtain:

$$a + \tanh^2(ax)\big(1 - a\big) = b + \tanh^2(x)\big(b^2 - b\big) \Rightarrow \tanh^2(x)\big(1 - \alpha - b^2 + b\big) = b - a$$
$$\Rightarrow b = a \text{ and } b^2 - b + a - 1 = 0 \Rightarrow b = a \text{ and } b^2 = 1 \Rightarrow b = a \text{ and } b = \pm 1.$$

The above is true because if $b^2 - b + a - 1 \neq 0$, then one would conclude that $\tanh^2(ax) = $ constant. Therefore:

$$\phi_{\sigma,1}(a) = a, \quad a = \{-1, 1\} \text{ and } \phi_{\sigma,d}(\mathbf{PQ}) = \mathbf{PQ}, \quad \mathbf{PQ} : \text{ signed permutation matrices.} \quad (36)$$

$\square$

### A.7.2 Theoretical analysis of ScaleGMN

**Proposition A.3** (Proposition 5.1 restated)**.** *ScaleGMN is permutation/scale equivariant or permutation/scale invariant when using a readout with the same symmetries.*

*Proof.* We will prove this by induction. In particular, consider two equivalent parameterisations $\boldsymbol{\theta}', \boldsymbol{\theta}$ as in Eq. (3). This implies that vertex/edge features will have the same symmetries, i.e. starting with the hidden neurons:

$$\mathbf{x}'_V(i) = q_\ell(\pi_\ell(i))\,\mathbf{x}_V(\pi_\ell(i)), \quad \ell = \text{layer}(i) \in \{1, \ldots, L - 1\}$$
$$\mathbf{x}_E(i, j) = q_\ell(\pi_\ell(i))\,\mathbf{x}_E(\pi_\ell(i), \pi_{\ell-1}(j))\,q_{\ell-1}^{-1}(\pi_{\ell-1}(j)), \quad \ell = \text{layer}(i) \in \{2, \ldots, L - 1\},$$

We will show that these symmetries are propagated to representations for every $t \geq 0$, i.e. that Eq. (4) and Eq. (5). Let us start with the base $t = 0$.

$$\mathbf{h}_V'^0(i) = \text{INIT}_V\left(\mathbf{x}_V'(i), \mathbf{p}_V(i)\right) = \text{INIT}_V\left(q_\ell(\pi_\ell(i))\,\mathbf{x}_V(\pi_\ell(i)), \mathbf{p}_V(i)\right)$$
$$= q_\ell(\pi_\ell(i))\,\text{INIT}_V\left(\mathbf{x}_V(\pi_\ell(i)), \mathbf{p}_V(i)\right) = q_\ell(\pi_\ell(i))\,\mathbf{h}_V^0(\pi_\ell(i)),$$
$$\mathbf{h}_E'^0(i) = \text{INIT}_E\left(\mathbf{x}_E'(i, j), \mathbf{p}_E(i, j)\right)$$
$$= \text{INIT}_E\left(q_\ell(\pi_\ell(i))\,\mathbf{x}_E(\pi_\ell(i), \pi_{\ell-1}(j))\,q_{\ell-1}^{-1}(\pi_{\ell-1}(j)), \mathbf{p}_E(i, j)\right)$$
$$= q_\ell(\pi_\ell(i))\,q_{\ell-1}^{-1}(\pi_{\ell-1}(j))\,\text{INIT}_E\left(\mathbf{x}_E(\pi_\ell(i), \pi_{\ell-1}(j)), \mathbf{p}_E(i, j)\right)$$
$$= q_\ell(\pi_\ell(i))\,q_{\ell-1}^{-1}(\pi_{\ell-1}(j))\,\mathbf{h}_E^0(\pi_\ell(i), \pi_{\ell-1}(j)).$$

This is due to the properties of the initialisation functions - see Eq. (20). Now we proceed with the induction step. Suppose Eq. (4) and Eq. (5) hold for $t-1$. We will show this implies they hold for $t$:

$$\mathbf{m}_V^{'t}(i) = \bigoplus_{j \in \mathcal{N}_{\text{FW}}(i)} \text{MSG}_V^t \left( \mathbf{h}_V^{'t-1}(i), \mathbf{h}_V^{'t-1}(j), \mathbf{h}_E^{'t-1}(i,j) \right)$$

$$= \bigoplus_{j \in \mathcal{N}_{\text{FW}}(i)} \text{MSG}_V^t \Big( q_\ell(\pi_\ell(i)) \mathbf{h}_V^{t-1}(\pi_\ell(i)), q_{\ell-1}(\pi_{\ell-1}(j)) \mathbf{h}_V^{t-1}(\pi_{\ell-1}(j)),$$

$$q_\ell(\pi_\ell(i)) \mathbf{h}_E^{t-1}(\pi_\ell(i), \pi_{\ell-1}(j)) q_{\ell-1}^{-1}(\pi_{\ell-1}(j)) \Big)$$

$$= \bigoplus_{j \in \mathcal{N}_{\text{FW}}(i)} q_\ell(\pi_\ell(i)) \text{MSG}_V^t \left( \mathbf{h}_V^{t-1}(\pi_\ell(i)), \mathbf{h}_V^{t-1}(\pi_{\ell-1}(j)), \mathbf{h}_E^{t-1}(\pi_\ell(i), \pi_{\ell-1}(j)) \right)$$

$$= q_\ell(\pi_\ell(i)) \bigoplus_{j \in \mathcal{N}_{\text{FW}}(i)} \text{MSG}_V^t \left( \mathbf{h}_V^{t-1}(\pi_\ell(i)), \mathbf{h}_V^{t-1}(j), \mathbf{h}_E^{t-1}(\pi_\ell(i), j) \right)$$

$$= q_\ell(\pi_\ell(i)) \mathbf{m}_V^t(\pi_\ell(i)).$$

$$\mathbf{h}_V^{'t}(i) = \text{UPD}_V^t \left( \mathbf{h}_V^{'t-1}(i), \mathbf{m}_V^{'t}(i) \right) = \text{UPD}_V^t \left( q_\ell(\pi_\ell(i)) \mathbf{h}_V^{t-1}(\pi_\ell(i)), q_\ell(\pi_\ell(i)) \mathbf{m}_V^t(\pi_\ell(i)) \right)$$

$$= q_\ell(\pi_\ell(i)) \text{UPD}_V^t \left( \mathbf{h}_V^{t-1}(\pi_\ell(i)), \mathbf{m}_V^t(\pi_\ell(i)) \right) = q_\ell(\pi_\ell(i)) \mathbf{h}_V^t(i)$$

$$\mathbf{h}_E^{'t}(i,j) = \text{UPD}_E^t \left( \mathbf{h}_V^{'t-1}(i), \mathbf{h}_V^{'t-1}(j), \mathbf{h}_E^{'t-1}(i,j) \right)$$

$$= \text{UPD}_E^t \Big( q_\ell(\pi_\ell(i)) \mathbf{h}_V^{t-1}(\pi_\ell(i)), q_{\ell-1}(\pi_{\ell-1}(j)) \mathbf{h}_V^{t-1}(\pi_{\ell-1}(j)),$$

$$q_\ell(\pi_\ell(i)) \mathbf{h}_E^{t-1}(\pi_\ell(i), \pi_{\ell-1}(j)) q_{\ell-1}^{-1}(\pi_{\ell-1}(j)) \Big)$$

$$= q_\ell(\pi_\ell(i)) q_{\ell-1}^{-1}(\pi_{\ell-1}(j)) \text{UPD}_E^t \left( \mathbf{h}_V^{t-1}(\pi_\ell(i)), \mathbf{h}_V^{t-1}(\pi_{\ell-1}(j)), \mathbf{h}_E^{t-1}(\pi_\ell(i), \pi_{\ell-1}(j)) \right)$$

$$= q_\ell(\pi_\ell(i)) q_{\ell-1}^{-1}(\pi_{\ell-1}(j)) \mathbf{h}_E^t(\pi_\ell(i), \pi_{\ell-1}(j)).$$

Therefore we have shown the desideratum for hidden neurons, where again we used the properties of the message and update functions (we omitted positional encodings here for brevity). Now for the input neurons, we have:

$$\mathbf{x}_V'(i) = \mathbf{x}_V(i), \quad \ell = \text{layer}(i) = 1$$

$$\mathbf{x}_E(i,j) = q_\ell(\pi_\ell(i)) \mathbf{x}_E(\pi_\ell(i), j), \ \ell = \text{layer}(i) = 1,$$

Further, we perform the induction again for Eq. (27) and Eq. (28) hold. Base case $t = 0$.

$$\mathbf{h}_V^{'0}(i) = \text{INIT}_{V,0} \left( \mathbf{x}_V'(i), \mathbf{p}_V(i) \right) = \text{INIT}_{V,0} \left( \mathbf{x}_V(i), \mathbf{p}_V(i) \right)$$

$$= \text{INIT}_{V,0} \left( \mathbf{x}_V(i), \mathbf{p}_V(i) \right) = \mathbf{h}_V^0(i).$$

Induction step:

$$\mathbf{m}_V^{'t}(i) = \mathbf{0} = \mathbf{m}_V^t(i).$$

$$\mathbf{h}_V^{'t}(i) = \text{UPD}_V^t \left( \mathbf{h}_V^{'t-1}(i), \mathbf{m}_V^{'t}(i) \right) = \text{UPD}_V^t \left( \mathbf{h}_V^{t-1}(i), \mathbf{m}_V^t(i) \right) = \mathbf{h}_V^t(i),$$

since input vertices do not have any incoming edge. Finally, for the output neurons, we have:

$$\mathbf{x}_V'(i) = \mathbf{x}_V(i), \quad \ell = \text{layer}(i) = L$$

$$\mathbf{x}_E(i,j) = q_{\ell-1}^{-1}(\pi_{\ell-1}(j)) \mathbf{x}_E(i, \pi_{\ell-1}(j)), \ \ell = \text{layer}(i) = L,$$

Induction for Eq. (27) and Eq. (29). Base case $t = 0$.

$$\mathbf{h}_V^{'0}(i) = \text{INIT}_{V,L} \left( \mathbf{x}_V'(i), \mathbf{p}_V(i) \right) = \text{INIT}_{V,L} \left( \mathbf{x}_V(i), \mathbf{p}_V(i) \right)$$

$$= \text{INIT}_{V,L} \left( \mathbf{x}_V(i), \mathbf{p}_V(i) \right) = \mathbf{h}_V^0(i),$$

$$\mathbf{h}_E^{'0}(i) = \text{INIT}_{E,L} \left( \mathbf{x}_E'(i,j), \mathbf{p}_E(i,j) \right)$$

$$= \text{INIT}_{E,L} \left( q_{\ell-1}^{-1}(\pi_{\ell-1}(j)) \mathbf{x}_E(i, \pi_{\ell-1}(j)), \mathbf{p}_E(i,j) \right)$$

$$= q_\ell^{-1}(\pi_{\ell-1}(j)) \text{INIT}_{E,L} \left( \mathbf{x}_E(i, \pi_{\ell-1}(j)), \mathbf{p}_E(i,j) \right)$$

$$= q_\ell^{-1}(\pi_{\ell-1}(j)) \mathbf{h}_E^0(i, \pi_{\ell-1}(j)).$$

And the induction step:

$$\mathbf{m}_V^{'t}(i) = \bigoplus_{j \in \mathcal{N}_{\mathrm{FW}}(i)} \mathrm{MSG}_V^t \left( \mathbf{h}_V^{'t-1}(i), \mathbf{h}_V^{'t-1}(j), \mathbf{h}_E^{'t-1}(i,j) \right)$$

$$= \bigoplus_{j \in \mathcal{N}_{\mathrm{FW}}(i)} \mathrm{MSG}_V^t \left( \mathbf{h}_V^{t-1}(i), q_{\ell-1}(\pi_{\ell-1}(j)) \mathbf{h}_V^{t-1}(\pi_{\ell-1}(j)), \right.$$

$$\left. \mathbf{h}_E^{t-1}(i, \pi_{\ell-1}(j)) q_{\ell-1}^{-1}(\pi_{\ell-1}(j)) \right)$$

$$= \bigoplus_{j \in \mathcal{N}_{\mathrm{FW}}(i)} \mathrm{MSG}_V^t \left( \mathbf{h}_V^{t-1}(i), \mathbf{h}_V^{t-1}(\pi_{\ell-1}(j)), \mathbf{h}_E^{t-1}(i, \pi_{\ell-1}(j)) \right)$$

$$= \bigoplus_{j \in \mathcal{N}_{\mathrm{FW}}(i)} \mathrm{MSG}_V^t \left( \mathbf{h}_V^{t-1}(i), \mathbf{h}_V^{t-1}(j), \mathbf{h}_E^{t-1}(i,j) \right)$$

$$= \mathbf{m}_V^t(i).$$

$$\mathbf{h}_V^{'t}(i) = \mathrm{UPD}_V^t \left( \mathbf{h}_V^{'t-1}(i), \mathbf{m}_V^{'t}(i) \right) = \mathrm{UPD}_V^t \left( \mathbf{h}_V^{t-1}(i), \mathbf{m}_V^t(i) \right) = \mathbf{h}_V^t(i)$$

$$\mathbf{h}_E^{'t}(i,j) = \mathrm{UPD}_E^t \left( \mathbf{h}_V^{'t-1}(i), \mathbf{h}_V^{'t-1}(j), \mathbf{h}_E^{'t-1}(i,j) \right)$$

$$= \mathrm{UPD}_E^t \left( \mathbf{h}_V^{t-1}(i), q_{\ell-1}(\pi_{\ell-1}(j)) \mathbf{h}_V^{t-1}(\pi_{\ell-1}(j)), \right.$$

$$\left. \mathbf{h}_E^{t-1}(i, \pi_{\ell-1}(j)) q_{\ell-1}^{-1}(\pi_{\ell-1}(j)) \right)$$

$$= q_{\ell-1}^{-1}(\pi_{\ell-1}(j)) \, \mathrm{UPD}_E^t \left( \mathbf{h}_V^{t-1}(i), \mathbf{h}_V^{t-1}(\pi_{\ell-1}(j)), \mathbf{h}_E^{t-1}(i, \pi_{\ell-1}(j)) \right)$$

$$= q_{\ell-1}^{-1}(\pi_{\ell-1}(j)) \, \mathbf{h}_E^t(i, \pi_{\ell-1}(j)).$$

Again we used the properties of the message and update functions we defined in Appendix A.1.4. And with that, we conclude the proof. $\qquad\square$

**Theorem A.4** (restated). *Consider a FFNN as per Eq. (1) with activation functions respecting the conditions of Proposition 4.1. Assume a Bidirectional-ScaleGMN with vertex update functions that can express the activation functions $\sigma_\ell$ and their derivatives $\sigma_\ell'$. Further, assume that ScaleGMN has access to the inputs $\mathbf{x}_0$ and the gradients of an (optional) loss function $\mathcal{L}$ w.r.t. to the output $\nabla_{\mathbf{x}_L} \mathcal{L}$, via its positional encodings. Then, ScaleGMN can simulate both a forward and a backward pass of the FFNN, by storing pre-activations, post-activations and their gradients at the vertex representations. In particular, to compute the forward pass at the FFNN layer $\ell$, $t \geq \ell$ ScaleGMN layers are required, while for the corresponding backward, the requirement is $t \geq 2L - \ell$.*

*Proof.* Consider a loss function $\mathcal{L} : \mathbb{R}^{d_{\mathrm{out}}} \times \mathbb{R}^{d_{\mathrm{out}}} \to \mathbb{R}$, which is computed on the output of the FFNN $\mathcal{L}(\mathbf{x}_L, \cdot)$. The second argument is optional and is used when we have labelled data - we will omit it in our derivations for brevity. Our proof strategy is based on two observations:

*First*, the pre-activations of each layer (denoted here as $\mathbf{z}_\ell = \mathbf{W}_\ell \mathbf{x}_{\ell-1} + \mathbf{b}_\ell$) **have the same symmetries as the biases**. This is straightforward to see by induction:

$$\mathbf{z}_0' = \mathbf{x}_0 = \mathbf{z}_0, \quad \mathbf{x}_0' = \mathbf{x}_0 \qquad\qquad \text{(base case 1)}$$

$$\mathbf{z}_1' = \mathbf{A}_1 \mathbf{W}_1 \mathbf{x}_0 + \mathbf{A}_1 \mathbf{b}_1 = \mathbf{A}_1 \mathbf{z}_1, \quad \mathbf{x}_1' = \sigma_1(\mathbf{A}_1 \mathbf{z}_1) = \phi_{\sigma_1}(\mathbf{A}_1) \mathbf{x}_1 \qquad \text{(base case 2)}$$

$$\mathbf{z}_\ell' = \mathbf{A}_\ell \mathbf{W}_\ell \phi_{\sigma_{\ell-1}}(\mathbf{A}_{\ell-1}^{-1}) \mathbf{x}_{\ell-1}' + \mathbf{A}_\ell \mathbf{b}_\ell = \mathbf{A}_\ell \mathbf{W}_\ell \phi_{\sigma_{\ell-1}}(\mathbf{A}_{\ell-1}^{-1}) \phi_{\sigma_{\ell-1}}(\mathbf{A}_{\ell-1}) \mathbf{x}_{\ell-1} + \mathbf{A}_\ell \mathbf{b}_\ell$$

$$= \mathbf{A}_\ell \mathbf{z}_\ell, \quad \mathbf{x}_\ell' = \phi_{\sigma_\ell}(\mathbf{A}_\ell) \mathbf{x}_\ell \qquad\qquad \text{(induction step)}$$

*Second*, the gradients of $\mathcal{L}$ w.r.t. the pre-activations of each layer (denoted here as $\nabla_{\mathbf{z}_\ell} \mathcal{L}(\mathbf{x}_L) = \left(\frac{\partial \mathbf{z}_{\ell+1}}{\partial \mathbf{z}_\ell}\right)^\top \nabla_{\mathbf{z}_{\ell+1}} \mathcal{L}(\mathbf{x}_L)$, where the dependence of $\mathbf{x}_L$ on $\mathbf{z}_\ell$ is implied - sometimes we will omit the argument $\mathbf{x}_L$ for brevity ) **have the inverted + transpose symmetries of the biases**. This is due to the invariance of the output to the symmetries considered:

$$\mathbf{x}_L(\mathbf{z}_\ell') = \mathbf{x}_L(\mathbf{z}_\ell) \iff \mathcal{L}(\mathbf{x}_L(\mathbf{z}_\ell')) = \mathcal{L}(\mathbf{x}_L(\mathbf{z}_\ell))$$

$$\iff \nabla_{\mathbf{z}_\ell} \mathcal{L}(\mathbf{x}_L(\mathbf{z}_\ell')) = \nabla_{\mathbf{z}_\ell} \mathcal{L}(\mathbf{x}_L(\mathbf{z}_\ell))$$

$$\iff \left(\frac{\partial \mathbf{z}_\ell'}{\partial \mathbf{z}_\ell}\right)^\top \nabla_{\mathbf{z}_\ell'} \mathcal{L}(\mathbf{x}_L(\mathbf{z}_\ell')) = \nabla_{\mathbf{z}_\ell} \mathcal{L}(\mathbf{x}_L(\mathbf{z}_\ell))$$

$$\iff \mathbf{A}_\ell^\top \nabla_{\mathbf{z}_\ell'} \mathcal{L}(\mathbf{x}_L(\mathbf{z}_\ell')) = \nabla_{\mathbf{z}_\ell} \mathcal{L}(\mathbf{x}_L(\mathbf{z}_\ell)) \iff \nabla_{\mathbf{z}_\ell'} \mathcal{L} = \left(\mathbf{A}_\ell^\top\right)^{-1} \nabla_{\mathbf{z}_\ell} \mathcal{L},$$

and similarly $\nabla_{\mathbf{x}'_\ell}\mathcal{L} = \left(\phi_{\sigma_\ell}(\mathbf{A}_\ell)^\top\right)^{-1}\nabla_{\mathbf{x}_\ell}\mathcal{L}$.

Now recall that in the cases we consider in this work, $\phi_{\sigma_\ell}(\mathbf{A}) = \mathbf{A}$ and $\mathbf{A} = \mathbf{PQ}$. Therefore, given that $(\mathbf{P}^\top)^{-1} = \mathbf{P}$ and $(\mathbf{Q}^\top)^{-1} = \mathbf{Q}^{-1}$, we have:

$$\mathbf{z}'_\ell = \mathbf{P}_\ell\mathbf{Q}_\ell\mathbf{z}_\ell, \quad \mathbf{x}'_\ell = \mathbf{P}_\ell\mathbf{Q}_\ell\mathbf{x}_\ell, \quad \nabla_{\mathbf{z}'_\ell}\mathcal{L} = \mathbf{P}_\ell\mathbf{Q}_\ell^{-1}\nabla_{\mathbf{z}_\ell}\mathcal{L}, \quad \nabla_{\mathbf{x}'_\ell}\mathcal{L} = \mathbf{P}_\ell\mathbf{Q}_\ell^{-1}\nabla_{\mathbf{x}_\ell}\mathcal{L}. \tag{37}$$

The latter is what motivates us to enforce vertex representations to have the same symmetries with the biases: **Assuming sufficient expressive power of ScaleGMN, we can anticipate that vertex representations will be able to reconstruct (1) the pre- or post-activations and (2) the elementwise-inverse of the pre- or post-activation gradients.**

Now recall the formulas that we want to recover:

$$\mathbf{z}_\ell(i) = \sum_{j\in\mathcal{N}_F(i)}\mathbf{W}_\ell(i,j)\mathbf{x}_{\ell-1}(j) + \mathbf{b}_\ell(i) = \sum_{j\in\mathcal{N}_F(i)}\mathbf{W}_\ell(i,j)\sigma_{\ell-1}\big(\mathbf{z}_{\ell-1}(j)\big) + \mathbf{b}_\ell(i)$$

$$\mathbf{x}_\ell(i) = \sigma_\ell\left(\mathbf{z}_\ell(i)\right) = \sigma_\ell\left(\sum_{j\in\mathcal{N}_F(i)}\mathbf{W}_\ell(i,j)\mathbf{x}_{\ell-1}(j) + \mathbf{b}_\ell(i)\right)$$

$$\nabla_{\mathbf{z}_\ell(i)}\mathcal{L} = \left(\left(\frac{\partial\mathbf{z}_{\ell+1}}{\partial\mathbf{z}_\ell}\right)^\top\nabla_{\mathbf{z}_{\ell+1}}\mathcal{L}\right)(i) = \sigma'_\ell\big(\mathbf{z}_\ell(i)\big)\sum_{j\in\mathcal{N}_B(i)}\mathbf{W}_\ell(j,i)\nabla_{\mathbf{z}_{\ell+1}(j)}\mathcal{L}$$

$$\nabla_{\mathbf{x}_\ell(i)}\mathcal{L} = \left(\left(\frac{\partial\mathbf{z}_{\ell+1}}{\partial\mathbf{x}_\ell}\right)^\top\nabla_{\mathbf{z}_{\ell+1}}\mathcal{L}\right)(i) = \sum_{j\in\mathcal{N}_B(i)}\mathbf{W}_\ell(j,i)\nabla_{\mathbf{z}_{\ell+1}(j)}\mathcal{L}$$

Now let us proceed to the proof. We will show that there exists a parameterization of ScaleGMN, such that after $t = \ell$ layers of message-passing, all vertices $i$ with layer $(i) \leq \ell$ will have stored in a part of their representation the pre-/post-activations for one or more inputs to the datapoint NN. Additionally, we will show that after $t = 2L - \ell$ layers of message-passing, all vertices $i$ with layer $(i) \geq \ell$ will have stored in a part of their representation the pre-/post-activation gradients for one or more inputs to the datapoint NN. Therefore, $L$ layers of (forward) message-passing are needed to compute the output of the datapoint NN, and $2L$ layers of (bidirectional) message passing to calculate the gradients of the output/loss w.r.t. the input.

For reasons that will become clearer as we proceed with the proof, we construct the below parameterization of ScaleGMN. First off, recall the input vertex and edge representations are given by:

$$\mathbf{x}_V(i) = \begin{cases} 1, \text{layer}(i) = 0, \\ \mathbf{b}_{\text{layer}(i)}(\text{pos}(i)), \text{layer}(i) \in [L], \end{cases}$$

$$\mathbf{x}_E(i,j) = \mathbf{W}_{\text{layer}(i)}(\text{pos}(i),\text{pos}(j)), i > j \quad \mathbf{x}_E(j,i) = \mathbf{1}\oslash\mathbf{W}_{\text{layer}(j)}(\text{pos}(j),\text{pos}(i)), i > j$$

*Then, we use the vertex positional encodings to store the information of the input values and the gradients w.r.t the outputs:*

$$\mathbf{p}_V(i) = \begin{cases} [\mathbf{z}_0(\text{pos}(i)),\mathbf{x}_0(\text{pos}(i))], \text{layer}(i) = 0 \\ [0,0], \text{layer}(i) \in [L-1] \\ [(\nabla_{\mathbf{z}_L(\text{pos}(i))}\mathcal{L})^{-1},(\nabla_{\mathbf{x}_L(\text{pos}(i))}\mathcal{L})^{-1}], \text{layer}(i) = L \end{cases}$$

*Throughout the network, the vertex representations will be 5-dimensional: we will use one to relay its neuron's bias across ScaleGMN's layers, two for the pre-and post-activations and two for their respective gradients. One can repeat the process for $m$ inputs, in which case the dimension will be $m + 1$. The edge representations will be 1-dimensional and will be simply used to relay the NN's weights throughout the network.*

Now, let us define the vertex initialisation functions.

$$\text{INIT}_{V,0}(\mathbf{x},\mathbf{p}_x) = \text{MLP}(\mathbf{x},\mathbf{p}) = [\mathbf{x}[1],\mathbf{p}_x[1],\mathbf{p}_x[2],0,0],$$
$$\text{INIT}_{V,L}(\mathbf{x},\mathbf{p}_x) = \text{MLP}(\mathbf{x},\mathbf{p}_x) = [\mathbf{x}[1],0,0,\mathbf{p}_x[1],\mathbf{p}_x[2]]$$
$$\text{INIT}_V(\mathbf{x},\mathbf{p}) = \text{AugScaleEq}(\mathbf{x},\mathbf{p}) = [\mathbf{x}[1],0,0,0,0].$$

Regarding the first two functions, it is obvious that can be expressed by an MLP. Regarding the third one, it can be expressed by AugScaleEq as follows: $\mathsf{AugScaleEq}(\mathbf{x},\mathbf{p}) = \mathbf{\Gamma}_x\mathbf{x}\odot \mathsf{AugScaleInv}(\mathbf{x},\mathbf{p}) = [1,0,0,0,0]^\top\mathbf{x}[1]\odot[1,1,1,1,1]^\top$, where the the constant vector can be expressed by AugScaleInv since it is invariant to the first argument. Further, for all edge initialisation and update functions (forward and backward):

$$\mathrm{INIT}_{E,*}^t(\mathbf{e},\mathbf{p}_e) = \mathbf{e}, \quad \mathrm{UPD}_{E,*}^t(\mathbf{x},\mathbf{y},\mathbf{e}) = \mathbf{e}$$

which can both be expressed by AugScaleEq using the identity as linear transform and an AugScaleInv with constant output.

We will not use positional encodings in the message and update functions, therefore we use their definitions of Eq. (8), Eq. (9). In particular, the forward/backward message functions:

$$\mathrm{MSG}_{V,\mathrm{FW}}^t\left(\mathbf{x},\mathbf{y},\mathbf{e}\right) = \mathsf{ScaleEq}\left([\mathbf{x},\mathsf{ReScaleEq}\left(\mathbf{y},\mathbf{e}\right)]\right) = \mathbf{y}[3]\cdot\mathbf{e}[1]$$

$$\mathrm{MSG}_{V,\mathrm{BW}}^t\left(\mathbf{x},\mathbf{y},\mathbf{e}\right) = g_m(\mathbf{y}[4]\cdot\mathbf{e}[1]),$$

where $g(\cdot) = \mathsf{ScaleEq}(\cdot)$ an arbitrarily expressive scale equivariant function. In the first case, this can be expressed as follows: zero linear transform on $\mathbf{x}$ (since it's unnecessary), identity transform on the output of ReScaleEq and a ScaleInv with constant output (ones). Then what remains is $\mathsf{ReScaleEq}\left(\mathbf{y},\mathbf{e}\right) = \mathbf{\Gamma}_y\mathbf{y}\odot\mathbf{\Gamma}_e\mathbf{e} = [0,0,1,0,0]^\top\mathbf{y}\cdot\mathbf{e}[1] = \mathbf{y}[3]\cdot\mathbf{e}[1]$.

Finally, the vertex update functions:

$$\mathrm{UPD}_{V,0}^t(\mathbf{x},\mathbf{m}_{\mathrm{FW}},\mathbf{m}_{\mathrm{BW}}) = \mathrm{MLP}(\mathbf{x},\mathbf{m}_{\mathrm{FW}},\mathbf{m}_{\mathrm{BW}})$$
$$= \left[\mathbf{x}[1],\mathbf{x}[2],\mathbf{x}[3],(\sigma_0'(\mathbf{x}[2])^{-1}\cdot g_U(\mathbf{m}_{\mathrm{BW}}[1]),g_U(\mathbf{m}_{\mathrm{BW}}[1])\right]$$
$$\mathrm{UPD}_{V,L}^t(\mathbf{x},\mathbf{m}_{\mathrm{FW}},\mathbf{m}_{\mathrm{BW}}) = \mathrm{MLP}(\mathbf{x},\mathbf{m}_{\mathrm{FW}},\mathbf{m}_{\mathrm{BW}})$$
$$= [\mathbf{x}[1],\mathbf{x}[1]+\mathbf{m}_{\mathrm{FW}}[1],\sigma_L\left(\mathbf{x}[1]+\mathbf{m}_{\mathrm{FW}}[1]\right),\mathbf{x}[4],\mathbf{x}[5]]$$
$$\mathrm{UPD}_V^t(\mathbf{x},\mathbf{m}_{\mathrm{FW}},\mathbf{m}_{\mathrm{BW}}) = \mathsf{ScaleEq}([\mathbf{x},\mathbf{m}_{\mathrm{FW}},\mathbf{m}_{\mathrm{BW}}])$$
$$= \left[\mathbf{x}[1],\mathbf{x}[1]+\mathbf{m}_{\mathrm{FW}}[1],\sigma\left(\mathbf{x}[1]+\mathbf{m}_{\mathrm{FW}}[1]\right),\right.$$
$$\left.(\sigma'(\mathbf{x}[2])^{-1}\cdot g_U(\mathbf{m}_{\mathrm{BW}}[1]),g_U(\mathbf{m}_{\mathrm{BW}}[1])\right],$$

where $g_U(\cdot) = \mathsf{ScaleEq}(\cdot)$ an arbitrarily expressive scale equivariant function. The first two can obviously be expressed by an MLP (note also that usually $\sigma_0,\sigma_L$ are the identity). Regarding the third, first, we assumed a common activation function $\sigma$ across hidden layers. In case the activation is not shared, a different update function should be used per layer. This is because, as we will see below, the activation and its derivative need to be learned for ScaleGMN to simulate the forward and the backward passes. Now, let's see how its element can be expressed by ScaleEq. Recall that $\mathsf{ScaleEq}(\mathbf{a}) = \mathbf{\Gamma}\mathbf{a}\odot\mathsf{ScaleInv}(\mathbf{a})$. Here, $\mathbf{\Gamma}\in\mathbb{R}^{5\times 7}$ (7 inputs, 5 outputs). The *first element (which will be the bias)*, the *second (pre-activation)* can be expressed by a ScaleInv with constant output (ones) and $\mathbf{\Gamma}(1)$ an all-zero vector except for the *1st* or the *1st* and *6th* coordinates respectively.

The *third (post-activation)*, *fourth (pre-activation gradient)* and the *fifth (post-activation gradient)* elements are not straightforward to simulate with elementary operations. However, one can observe that all are scale equivariant. In particular, the third: $\sigma(qx+qy) = q\sigma(x+y)$ (by assumption about the symmetries of the activation function). Additionally, the fourth (assuming a scale equivariant $g_U$): $\frac{g_U(qy)}{\sigma'(qx)} = \frac{q\cdot g_U(y)}{\sigma'(x)}$ - this is because the derivative of a *scale equivariant* activation function is *scale invariant*.[8] The fifth is by definition scale equivariant. *Therefore, if* ScaleEq *is sufficiently expressive, so as to contain all scale equivariant functions*, then it can express the third to fifth elements.

It remains to establish a few last statements that will be necessary in our induction. *First, the edge representations are constant throughout ScaleGMN and equal to the weights of the input NN*:

$$\mathbf{h}_E^0(i,j) = \mathrm{INIT}_{E,*}^t\left(\mathbf{x}_E(i,j),\mathbf{p}_E(i,j)\right) = \mathbf{x}_E(i,j) = \mathbf{W}_{\mathrm{layer}(i)}(\mathrm{pos}(i),\mathrm{pos}(j))$$
$$\mathbf{h}_E^t(i,j) = \mathrm{UPD}_{E,*}^t\left(\mathbf{h}_V^{t-1}(i),\mathbf{h}_V^{t-1}(j),\mathbf{h}_E^{t-1}(i,j)\right) = \mathbf{h}_E^{t-1}(i,j) = \cdots = \mathbf{h}_E^0(i,j).$$

---

[8] $\sigma(qx) = q\sigma(x) \implies q\sigma'(qx) = q\sigma'(x) \implies \sigma'(qx) = \sigma'(x)$ for non-zero scalar multipliers.

*Second, the first element of the vertex representations is constant throughout ScaleGMN and equal to the biases of the input NN:*

$$\mathbf{h}_V^0(i)[1] = \text{INIT}_{V,*}^t\left(\mathbf{x}_V(i), \mathbf{p}_V(i)\right)[1] = \mathbf{x}_V(i)[1] = \begin{cases} 1, \text{layer}(i) = 0, \\ \mathbf{b}_{\text{layer}(i)}(\text{pos}(i)), \text{layer}(i) \in [L]. \end{cases}$$

$$\mathbf{h}_V^t(i)[1] = \text{UPD}_{V,*}^t\left(\mathbf{h}_V^{t-1}(i), \mathbf{m}_{V,\text{FW}}^t(i), \mathbf{m}_{V,\text{BW}}^t(i)\right) = \mathbf{h}_V^{t-1}(i)[1] = \cdots = \mathbf{h}_V^0(i)[1]$$

Now, our *first induction hypothesis is the following: If for all vertices with layer$(i) = \ell$ and $t \geq \ell$ we have that $\mathbf{h}_V^t(i)[2:3] = [\mathbf{z}_\ell(pos(i)), \mathbf{x}_\ell(pos(i))]$, then the same should hold for vertices with layer$(i) = \ell + 1$ when $t \geq \ell + 1$.*

- (*Base case:*) If layer$(i) = 0$:

$$\mathbf{h}_V^t(i)[2:3] = \text{UPD}_{V,0}^t\left(\mathbf{h}_V^{t-1}(i), \mathbf{m}_V^t(i)\right)[2:3]$$
$$= \mathbf{h}_V^{t-1}(i)[2:3] = \cdots = \mathbf{h}_V^0(i)[2:3]$$
$$= \text{INIT}_{V,0}\left(\mathbf{x}_V(i), \mathbf{p}_V(i)\right)[2:3]$$
$$= [\mathbf{z}_0(\text{pos}(i)), \mathbf{x}_0(\text{pos}(i))]$$

Therefore, the hypothesis holds for $t \geq 0 = \text{layer}(i)$ for the base case.

- (*Induction step:*) Suppose the hypothesis holds for vertices with layer$(i) = \ell - 1 < L$. Now, if layer$(i) = \ell$, for $t \geq \ell$ we have:

$$\mathbf{m}_{V,\text{FW}}^t(i) = \bigoplus_{j \in \mathcal{N}_F(i)} \text{MSG}_{V,\text{FW}}^t\left(\mathbf{h}_V^{t-1}(i), \mathbf{h}_V^{t-1}(j), \mathbf{h}_E^{t-1}(i,j)\right)$$
$$= \sum_{j \in \mathcal{N}_F(i)} \mathbf{h}_V^{t-1}(j)[3] \cdot \mathbf{h}_E^{t-1}(i,j)[1] \quad (\text{layer}(j) = \ell - 1, t - 1 \geq \ell - 1)$$
$$= \sum_{j \in \mathcal{N}_F(i)} \mathbf{x}_{\ell-1}(\text{pos}(j)) \cdot \mathbf{W}_\ell(\text{pos}(i), \text{pos}(j))$$
$$\mathbf{h}_V^t(i)[2:3] = \text{UPD}_V^t\left(\mathbf{h}_V^{t-1}(i), \mathbf{m}_V^t(i)\right)[2:3]$$
$$= \left[\mathbf{h}_V^{t-1}(i)[1] + \mathbf{m}_{V,\text{FW}}^t(i), \sigma\left(\mathbf{h}_V^{t-1}(i)[1] + \mathbf{m}_{V,\text{FW}}^t(i)\right)\right]$$
$$= \Big[\mathbf{b}_\ell(\text{pos}(i)) + \sum_{j \in \mathcal{N}_F(i)} \mathbf{x}_{\ell-1}(\text{pos}(j)) \cdot \mathbf{W}_\ell(\text{pos}(i), \text{pos}(j)),$$
$$\sigma_\ell\left(\mathbf{h}_V^{t-1}(i)[1] + \mathbf{m}_{V,\text{FW}}^t(i)\right)\Big]$$
$$= [\mathbf{z}_\ell(\text{pos}(i)), \sigma_\ell(\mathbf{z}_\ell(\text{pos}(i)))] = [\mathbf{z}_\ell(\text{pos}(i)), \mathbf{x}_\ell(\text{pos}(i))].$$

Similarly for the vertices with layer$(i) = L$

Our *second induction hypothesis is the following: If for all vertices with layer$(i) = \ell$ and $t \geq 2L - \ell$ we have that $\mathbf{h}_V^t(i)[4:5] = \left[(\nabla_{\mathbf{z}_\ell(pos(i))}\mathcal{L})^{-1}, (\nabla_{\mathbf{x}_\ell(pos(i))}\mathcal{L})^{-1}\right]$, then the same should hold for vertices with layer$(i) = \ell - 1$ when $t \geq 2L - \ell + 1$.*

- (*Base case:*) If layer$(i) = L$:

$$\mathbf{h}_V^t(i)[4:5] = \text{UPD}_{V,L}^t\left(\mathbf{h}_V^{t-1}(i), \mathbf{m}_V^t(i)\right)[4:5]$$
$$= \mathbf{h}_V^{t-1}(i)[4:5] = \cdots = \mathbf{h}_V^0(i)[4:5]$$
$$= \text{INIT}_{V,L}\left(\mathbf{x}_V(i), \mathbf{p}_V(i)\right)[4:5]$$
$$= \left[(\nabla_{\mathbf{z}_L(\text{pos}(i))}\mathcal{L})^{-1}, (\nabla_{\mathbf{x}_L(\text{pos}(i))}\mathcal{L})^{-1}\right]$$

Therefore, the hypothesis holds for $t \geq L = \text{layer}(i)$ for the base case.

- (*Induction step*:) Suppose the hypothesis holds for vertices with layer $(i) = \ell + 1$. Now, if layer $(i) = \ell$, for $t \geq 2L - \ell$ we have:

$$\mathbf{m}_{V,\text{BW}}^t(i) = \bigoplus_{j \in \mathcal{N}_B(i)} \text{MSG}_{V,\text{BW}}^t \left( \mathbf{h}_V^{t-1}(i), \mathbf{h}_V^{t-1}(j), \mathbf{h}_E^{t-1}(i,j) \right)$$

$$= \sum_{j \in \mathcal{N}_B(i)} g_m(\mathbf{h}_V^{t-1}(j)[4] \cdot \mathbf{h}_E^{t-1}(i,j)[1]) \; (t - 1 \geq 2L - (\ell + 1))$$

$$= \sum_{j \in \mathcal{N}_B(i)} g_m \left( (\nabla_{\mathbf{z}_{\ell+1}(\text{pos}(j))} \mathcal{L})^{-1} \cdot \mathbf{W}_\ell^{-1}(\text{pos}(j), \text{pos}(i)) \right).$$

$$\mathbf{h}_V^t(i)[4:5] = \text{UPD}_V^t \left( \mathbf{h}_V^{t-1}(i), \mathbf{m}_V^t(i) \right)[4:5]$$

$$= \left[ \frac{g_U \left( \mathbf{m}_{V,\text{BW}}^t(i) \right)}{\sigma_\ell' \left( \mathbf{h}_V^{t-1}(i)[2] \right)}, g_U \left( \mathbf{m}_{V,\text{BW}}^t(i) \right) \right]$$

$$= \left[ \frac{1}{\sigma_\ell'(\mathbf{z}_\ell(\text{pos}(i)) \nabla_{\mathbf{x}_\ell(\text{pos}(i))} \mathcal{L}}, (\nabla_{\mathbf{x}_\ell(\text{pos}(i))} \mathcal{L})^{-1} \right]$$

$$= \left[ (\nabla_{\mathbf{z}_\ell(\text{pos}(i))} \mathcal{L})^{-1}, (\nabla_{\mathbf{x}_\ell(\text{pos}(i))} \mathcal{L})^{-1} \right].$$

In the last step, we assumed the following: $g_U \left( \sum g_m(x) \right)$ is a sufficiently expressive scale equivariant and permutation invariant function. This is in order to express the following:

$$(\nabla_{\mathbf{x}_\ell(\text{pos}(i))} \mathcal{L})^{-1} = \left( \sum_{j \in \mathcal{N}_B(i)} \nabla_{\mathbf{z}_{\ell+1}(\text{pos}(j))} \mathcal{L} \cdot \mathbf{W}_\ell(\text{pos}(j), \text{pos}(i)) \right)^{-1} \text{ with:}$$

$$(\nabla_{q_\ell(i) \mathbf{x}_\ell(\text{pos}(i))} \mathcal{L})^{-1} = \left( \sum q_{\ell+1}(j)^{-1} \nabla_{\mathbf{z}_{\ell+1}(\text{pos}(j))} \mathcal{L} \cdot \frac{q_{\ell+1}(j)}{q_\ell(i)} \mathbf{W}_\ell(\text{pos}(j), \text{pos}(i)) \right)^{-1}$$

$$= \frac{q_\ell(i)}{\sum \nabla_{\mathbf{z}_{\ell+1}(\text{pos}(j))} \mathcal{L} \cdot \mathbf{W}_\ell(\text{pos}(j), \text{pos}(i)))}$$

$$= q_\ell(i)(\nabla_{\mathbf{x}_\ell(\text{pos}(i))} \mathcal{L})^{-1}.$$

Therefore if all $g_U \left( \sum g_m(qx) \right) = q \cdot g_U \left( \sum g_m(x) \right)$ can be expressed by $g_U, g_m$, then so is the inverse of the gradient.

Following a similar rationale we can prove the case for the vertices with layer $(i) = 0$.

$\square$

