# OpenReview forum: "Scale Equivariant Graph Metanetworks"
_NeurIPS.cc/2024/Conference — NeurIPS 2024 oral_

### Official Review · Reviewer_ugM4 · 2024-07-02

**Soundness:** 3
**Presentation:** 2
**Contribution:** 3
**Rating:** 7
**Confidence:** 4

**Summary:**

The paper considers the emergent and fascinating field of learning over weight spaces, that is, neural nets that process other neural networks. The processing NN is referred to as a metanetwork. Previous approaches showcased the importance of accounting for the input NN’s symmetries by designing equivariant architectures. However, they were mainly focused on permutation symmetries. This paper proposes a GNN-based metanetwork, which is permutation and scale equivariant. The paper studies the expressiveness (in terms of simulating the forward and backward pass of the input NN) of the proposed arch. The proposed method is evaluated using several INRs and classifiers datasets.

**Strengths:**

1. The paper deals with an important and timely problem of learning in deep weight spaces, and presents a novel architecture by incorporating scale and permutation equivariance.
2. The paper provides theoretical analysis and results regarding the expressive power of the proposed approach.
3. Empirical results show significant improvement over baseline methods.

**Weaknesses:**

My main concern is the limited empirical evaluation and missing natural baselines. While the presented empirical results show significant improvement over baseline methods, at the current state of the learning over weight spaces literature, I would expect a more diverse, challenging, and comprehensive empirical evaluation.

1. The writing and formatting require enhancement and refinement—specifically, long sentences, many slashes, many footnotes, etc. Also, in my view, the proposed method is introduced too late in the paper (page 6).
2. A missing natural baseline is to use a permutation equivariant baseline like DWS/NFN or NG-GNN together with scaling data augmentations as in [1].
3. The experimental section only considers invariant tasks. Additional experiments using equivariant tasks (e.g. INR editing, domain adaptation, etc.) would significantly strengthen the empirical study.
4. Some evaluation and comparison of runtime and memory consumption w.r.t. baselines would be beneficial.
5. Furthermore, adding experiments with larger input networks and diverse input architectures (like a varying number of layers) would again significantly strengthen the empirical study.
6. Also, adding some ablation regarding design choices would be beneficial.
7. Why are the results for DWS and INR2Vec missing in Table 1?
8. Minor:
   - Line 198: should be a^k.
   - Line 336: “transformations We” -> “transformations. We”

References:

[1] Improved Generalization of Weight Space Networks via Augmentations, ICML 2024.

**Questions:**

1. Is it always feasible and relatively easy to design and implement either the canon or symm mappings for all activation functions?
2. The bidirectional version of the method achieves on-par performance as the non-bidirectional one, except for the Augmented CIFAR-10 dataset, where the performance is much worse. Could you provide some insights regarding this result?
3. Additionally, how does the ScaleGMNB and ScaleGMN compare in terms of runtime?
4. How do ScaleGMN and ScaleGMNB compare to GMN in terms of the number of trainable parameters and runtime?
5. Did you use any augmentation on the input NNs?
6. Are there any limitations on the choice of activations of the ScaleGMN network?

**Limitations:**

Yes.

---

> ### Author Rebuttal · Authors · 2024-08-06
>
> *For weaknesses: 1,3,4,5,7 and question 3 please refer to Comment.*
>
> ### Weaknesses
>
> >*Weakness 2. Scaling data augmentations*
>
> An effective method to get such augmentations is to sample scaling matrices for every training datapoint  - diagonal sign matrices for sign and diagonal positive scaling matrices for positive scale symmetries, respectively. Subsequently, we transform the input network parameters by applying the matrices to the hidden layers. (like in Eq.3 in our paper, but without the permutation matrices)
>
> The way to perform this random sampling, however, is not straightforward and we have to deal the two cases separately.
>
> **Sign symmetries**: Since the diagonal sign matrices are of the form $\mathbf{Q} = \text{diag}(q_1 , \dots , q_d ), q_i = ±1$, we sample each element of the matrix **independently and uniformly at random with probability 0.5**. We observe that augmenting the training set leads consistently to better results when compared to the original baselines. **None of these methods however achieved results on par with ScaleGMN and ScaleGMN-B.** (Table 3 in the PDF)
>
> **Positive-scale symmetries**: The diagonal positive scaling matrices are of the form: $\mathbf{Q} = \text{diag}(q_1 , \dots , q_d ), q_i \in (0, +\infty)$. Hence, we have to sample from a **continuous and unbounded distribution** making the augmentation strategy much more difficult to design. Consulting the plots in the appendix A.5, we opted for an exponential distribution and experiment with the coefficient $\lambda$. Nevertheless, regardless of the distribution choice we cannot guarantee that the augmentations will be sufficient, due to the lack of upper bound. We observe that we were not able to surpass the original baselines, which indicates that designing an effective baseline of this type is not straightforward. (Table 3 in the PDF)
>
> We thank the reviewer for proposing this important baseline.
>
> ### Questions
>
> >*Question 1. Canon/symm mappings for all activation functions.*
>
> In general, we cannot guarantee that these mappings are easy to construct for *any* activation function, since it is currently unknown if we can provide a general characterisation of the possible symmetries that may arise. Our results  extend to all positively homogeneous activations, $\sigma(\lambda x) = \lambda \sigma(x), \lambda > 0$, and all odd ones, $\sigma(-x) = - \sigma(x)$. We refer the reviewer to Table 1 of [1], where we can see that LeakyReLU also falls in the first category. Regarding polynomial activations, which demonstrate *non-zero scaling symmetries*, one option would be: (1) norm division to canonicalise scale, and (2) sign symm/canon as discussed in the main paper (and the rebuttal). The above, cover a fairly broad spectrum of common activation functions.
>
>
> >*Question 2. Forward vs Bidirectional and Augmented CIFAR-10.*
>
> Indeed, in most invariant tasks, the bidirectional variant does not provide significant advantages. This is inline with what has been observed in [2], where a forward variant is also sufficient. We speculate that this might be related to the fact that the tasks we considered may be solved by simulating the forward pass of the input NN alone. Given that the forward pass can be expressed by the forward variant alone, this might provide an explanation for its efficacy on the invariant tasks we considered. The significance of the bidirectional variant is highlighted mostly on  *equivariant* tasks, where ScaleGMN-B significantly outperforms the forward variant. Please refer to the global response, where we report our results on INR-editing and discuss the limitations of the forward variant on the equivariant tasks.
>
> Please refer to our response to Question 4 of reviewer x2pG for ScaleGMN-B on the Augmented CIFAR-10.
>
>
> >*Question 4. Trainable parameters: GMN vs ScaleGMN(-B)?*
>
> Going from a GMN model to ScaleGMN requires adapting the MSG and UPD functions to be scale equivariant, leading to more learnable parameters, as opposed to using plain MLPs. Another design choice of ScaleGMN that introduces more parameters is using different MLPs for the I/O nodes (Appendix A.1.4).
>
> Please refer to Table 2, where we report the training and inference runtimes.
>
>
> >*Question 5. Did you use any augmentation on the input NNs?*
> *No, we do not use any augmentation on the input NNs.*
>
> Augmentations are only used for the dataset "Augmented CIFAR-10", where we follow the augmentation procedure from [3] for comparison reasons with the rest of the baselines. ScaleGMN and ScaleGMN-B rely solely on the original training dataset and on built-in equivariance/invariance.
>
> >*Question 6. Are there any limitations on the choice of activations of the ScaleGMN network?*
>
> *Importantly, our method does **not** impose any limitations on the choice of the activation functions.*
>
> We are able to select any activation, because these are only applied within the MLPs of the invariant modules.  As discussed in Section 5 of our paper, the MLPs (equipped with non-linearities) are only applied after the canon/symm function. In case one chose to place activations in a different computational part of ScaleGMN, this would indeed limit their options so as not to compromise scale equivariance. However, this is not the case in our method. We thank the reviewer for noticing this important detail.
>
> ---
> [1] Godfrey, Charles, et al. "On the symmetries of deep learning models and their internal representations." Advances in Neural Information Processing Systems 35 (2022): 11893-11905.
>
> [2] Kofinas, Miltiadis, et al. "Graph Neural Networks for Learning Equivariant Representations of Neural Networks." The Twelfth International Conference on Learning Representations.
>
> [3] Zhou, Allan, et al. "Permutation equivariant neural functionals." Advances in neural information processing systems 36 (2024).
>
> [4] Shamsian, Aviv, et al. "Improved generalization of weight space networks via augmentations." arXiv preprint arXiv:2402.04081 (2024).

---

> ### Author Response · Authors · 2024-08-06
> **Additional responses to reviewer Reviewer ugM4**
>
> >*Weakness 1. Writing.*
>
> Please refer to our global response regarding the structure of our paper and the writing style.
>
> >*Weakness 3. Equivariant tasks.*
>
> Please refer to our global response, where we provide details regarding the INR editing task as well as to Table 1 of the attached PDF.
>
>
> >*Weakness 4. Runtime and memory consumption.*
>
> We thank the reviewer for suggesting a comparison of runtime and memory consumption. We select the F-MNIST dataset and make two comparisons; at first we report the number of parameters of all the reported models. Regarding the runtime, we fix the number of parameters for fairness of comparison and report the training time, inference time and GPU memory consumption. Please refer to Table 2 of the attached PDF. We can see that ScaleGMN-B does not introduce performance degradation regarding the runtime, while both our methods are quiet slower than the baselines. Since we do not use computationally much heavier operations, this last result indicates that our implementation could be further optimized w.r.t. runtime.
>
>
> >*Weakness 5. Larger and diverse input architectures.*
>
> Please refer to our global response regarding the datasets and experiments on larger and more complex architectures.
>
>
> >*Weakness 7. DWS and INR2VEC on CIFAR-10*
>
> Please refer to our response to reviewer x2pG regarding these two experiments.
>
> >*Question 3. Runtime performance of ScaleGMN and ScaleGMN-B.*
>
> Please refer to Table 2 of the attached PDF. As discussed in *Weakness 4*, ScaleGMN-B does not introduce performance degradation regarding the runtime.

---

> > ### Comment · Reviewer_ugM4 · 2024-08-11
> > **Response to rebuttal**
> >
> > I would like to thank the authors for their rebuttal and for providing additional results and discussion, which addresses some of my concerns. I've read all the reviewers' concerns and the authors' responses. The authors addressed many of my concerns regarding the limited and missing empirical evaluation, so I raised my score accordingly.

---

### Official Review · Reviewer_x2pG · 2024-07-03

**Soundness:** 3
**Presentation:** 3
**Contribution:** 3
**Rating:** 6
**Confidence:** 4

**Summary:**

This paper addresses the emerging and fascinating field of deep-weight space learning where neural nets used to process weights and biases of another deep model. The authors have introduced new methods based on GNN architecture called ScaleGMN and ScaleGMNB for Scale Equivariant Graph MetaNetworks. The latter is a bi-directional variant. Both approaches tackle the scale symmetries presented by the input neural model's activation functions.

The authors claim the following contributions:
- Extending the scope of metanetwork design from permutation to scaling symmetries.
- Designing networks that are invariant or equivariant to scalar multiplication from arbitrary scaling groups.
- Theoretical analysis of the expressive power of ScaleGMN.
- Extensive empirical comparison with recent work on various datasets in the field of weight space learning.

**Strengths:**

- With the growing number of works on Implicit Neural Representation (INR) and the increasing need to process neural networks, this paper tackles a crucial field and problem, advancing the area in a way that can benefit many practitioners.
- The authors introduce a new structure that incorporates both permutation and scale symmetries by ensuring it is equivariant to scale and permutation.
- The authors provide a theoretical analysis of the expressive power of the proposed approaches. Additionally, they show that ScaleGMN can simulate forward and backward passes of arbitrary inputs in the processed NN.
- The empirical results show a significant improvement compared to recent works in the field of weight space learning.

**Weaknesses:**

- The writing can be significantly improved, particularly by breaking down long sentences that are hard to follow. Additionally, the writing pace is somewhat slow. While this might be beneficial for the average reader, the authors allocate too much space to exposition and problem formulation. As a result, the presentation of the proposed methods begins relatively late in the paper.
- The experimental section focuses only on invariant tasks, i.e. INR classification and NN generalization prediction. It would be interesting to see how well ScaleGMN and ScaleGMNB deal with equivariant tasks like the ones presented in [1,2] which are considered harder for weight space architectures.
- The processed architectures are not diverse dealing only with small-sized feed-forward and CNN architectures. It would be interesting to see more diversity in the processed architectures like deeper nets, attention-based methods, etc.

-------
[1] Equivariant Architectures for Learning in Deep Weight Spaces, Navon et al.

[2] Permutation Equivariant Neural Functionals, Zhou et al.

**Questions:**

- Why DWS and INR2VEC are missing in Table 1? (CIFAR-10 | Augmented CIFAR-10 experiments).
- In common bi-directional architectures (e.g. LSTM) we see performance (runtime) degradation compared to non-bi-directional design is that the same for ScaleGMN and ScaleGMNB? What is the computational complexity of both methods?
- Can ScaleGMN and ScaleGMNB handle input data with heterogeneous activation functions, i.e. one network with ReLU activations and another with only tanh activations? (as long as they respect Prop. 4.1)
- The results for the Augmented CIFAR-10 experiments are odd. ScaleGMNB performs worse than all baselines except MLP, while in other experiments, it outperforms them. Do the authors have an explanation for this observation?
- Adding a figure that illustrates ScaleGMN and ScaleGMNB architectures and their design w.r.t permutation and scale symmetries, would be beneficial.

**Limitations:**

See above.

---

> ### Author Rebuttal · Authors · 2024-08-06
>
> ### Weaknesses
>
> >*Weakness 1. Writing Style*
>
> We thank the reviewer for their suggestions. Please refer to the global response about the writing pace and style.
>
> >*Weakness 2. Equivariant tasks*
>
> To evaluate the performance of our method on tasks that require permutation and scale equivariance, we selected the task of INR editing. Please refer to our global response and to Table 1 of the attached PDF. In summary, the bidirectional variant, ScaleGMN-B, surpasses all baseline metanetworks, including GNNs using additional information (probe features)
>
> >*Weakness 3. Processing diverse and complex architectures*
>
> Please refer to our global response regarding the experiments on larger and more complex architectures.
>
> ### Questions
> >*Question 1.  DWS and INR2VEC on CIFAR-10*
>
>
> In Table 1 of our paper we include, together with our results, all the baselines from the literature. For the CIFAR-10 dataset we did not include any experiments with DWS and INR2VEC, as they were not included in [1], where the task was proposed. However, we have now run both experiments using the **DWS** model, which achieved $34.45 \pm 0.42$ on CIFAR-10 and $41.27 \pm 0.026$ on Augmented CIFAR-10. As expected, these results are on par with the rest of the permutation equivariant models.
>
> Regarding INR2VEC [2], we have not reimplemented it and tested on the CIFAR-10 dataset - we only used results present in the literature. However, INR2VEC demonstrated significantly worse performance than DWS [3] and NFN [1] on the task of INR classification, as can be found in the literature, and therefore testing it on extra data will probably offer limited added value, as it. Nevertheless, we plan to also include this baseline in an updated version.
>
>
> >*Question 2. Performance (runtime) degradation between ScaleGMN and ScaleGMNB and complexity*
>
>
> The reviewer here makes a correct comment regarding the complexity of the forward (ScaleGMN) and the bidirectional variant (ScaleGMN-B) of our method. In the latter case, we add *backward edges* to our graph and introduce a *backward message function* to discern the two message directions. Subsequently, we  concatenate the outputs of the two message functions and apply the UPD function. Consequently, the additional complexity of ScaleGMN-B is introduced solely by the extra message function, with a complexity of $O(E)$. Given that ScaleGMN has the complexity of a standard GNN model $O(V+E)$, the final complexity of ScaleGMN-B is $O(V+2E)$.
>
> To measure the differences regarding the runtime performance, we conduct a controlled experiment on the F-MNIST dataset. We observe that ScaleGMN-B only needs $6.83$ more seconds per epoch, when compared to ScaleGMN. Regarding the inference time this difference is $0,0132$ seconds per datapoint. Please refer to our response to reviewer ugM4 as well as to Table 2 in the PDF within our global response for the complete results.
>
>
> >*Question 3. Heterogeneous activation functions.*
>
>
> This is an interesting question and of importance to ensure the generality of our method. *In principle, our method does not impose any limitations regarding the homogeneity of the activation functions of the input neural networks*. To see this, observe that the only part of the metanetwork that gets affected by the symmetry induced by the activation function, is the *symmetrisation/canonicalisation* function. In other words, all the modules of the metanetwork can be reused for any input NN with arbitrary activations, as long as the datapoints are symmetrised/canonicalised accordingly.
>
> **Experiments**. Experimentally, we opted to split the datasets into two subsets (one with ReLU Nets and one with tanh Nets). This choice was made solely to evaluate our method separately for each type of symmetry and assess the different invariant methods that we employed for each case. Following, the reviewer's suggestion, we extend our evaluation to heterogeneous activation functions (a dataset containing both ReLU and tanh Nets). We conducted epxeriments on the CIFAR-10-GS dataset and report the results on Table 4 in the attached PDF of our global response. The baselines are reported as in [4]. **Interestingly, we observe that ScaleGMN demonstrates superior performance compared to the previous baselines, significantly exceeding the performance of the next best model.** We thank the reviewer for suggesting this experiment - we will include this in the updated version of the manuscript.
>
>
> >*Question 4. The results of ScaleGMN-B on Augmented CIFAR-10.*
>
>
> This was indeed a confusing result, which was merely due to suboptimal hyperparameter search. Unfortunately, due to the large size of this dataset ($20$ times larger than "CIFAR-10") and limited computational resources, the hyperparameter search for the ScaleGMN-B on the Augmented CIFAR-10 dataset had not finished by the time of the submission. Hence, the reported result does not reflect the real performance of the model. We completed the hyperparameter search post-submission, and achieved accuracy equal to **$56.95 \pm 0.57$** - this result follows the same pattern with the rest of the datasets.
>
>
> >*Question 5. Figure of ScaleGMN and ScaleGMNB.*
>
> We thank the reviewer for suggesting to include a figure depicting our architectures. We will consider designing one and including it in an updated version of the manuscript.
>
> ---
> [1] Zhou, Allan, et al. "Permutation equivariant neural functionals." Advances in neural information processing systems 36 (2024).
>
> [2] De Luigi, Luca, et al. "Deep Learning on Implicit Neural Representations of Shapes." The Eleventh International Conference on Learning Representations.
>
> [3] Navon, Aviv, et al. "Equivariant architectures for learning in deep weight spaces." International Conference on Machine Learning. PMLR, 2023.
>
> [4] Kofinas, Miltiadis, et al. "Graph Neural Networks for Learning Equivariant Representations of Neural Networks." The Twelfth International Conference on Learning Representations.

---

> > ### Comment · Reviewer_x2pG · 2024-08-12
> > **Reviewer response**
> >
> > I would like to express my gratitude to the authors for the time and effort they have dedicated to this rebuttal.
> > Most of my concerns have been addressed and after reading the other reviews and comments I decided to raise my score.

---

### Official Review · Reviewer_AXwX · 2024-07-05

**Soundness:** 4
**Presentation:** 4
**Contribution:** 4
**Rating:** 8
**Confidence:** 4

**Summary:**

This work develops new GNN-based metanetworks that are equivariant to scaling symmetries induced by nonlinearities in input neural networks. Their ScaleGMNs extend metanetworks, which are typically only permutation equivariant (if at all equivariant), to also account for other symmetries in input neural networks' parameters. The architecture is proved to be equivariant to the desired symmetries and also expressive in that it can simulate forward and backward passes of the input. Experiments show improvements over merely-permutation-equivariant metanetworks.

**Strengths:**

1. Great writing. Nice introduction and related work, as well as good setup and notation in Section 3.
2. Nice theoretical results. That ScaleGMN is can express the forward and backward pass is a good way to check that its expressive power is not overly limited when adding the additional scaling equivariances. Also, there is an interesting discussion in Appendix A.2 on equivariant for bidirectional ScaleGMNs.
3. Large empirical improvements, especially on INR classification (without many of the unfair or expensive tricks that others use!), with ScaleGMN. I say some previous tricks are "unfair" because, for instance, random probe features of Kofinas et al. as used on INRs can essentially be used to see the input image pixels, and hence the prediction task is also taking in the image as well as the INR representing it. That ScaleGMN can beat the prior methods without these tricks is very impressive.
4. Several other interesting empirical findings. These include the fact that ScaleGMN does not need random Fourier features or data augmentations, and that the bidirectional version can vary in performance (sometimes drastically as in augmented CIFAR-10).

**Weaknesses:**

1. It would be interesting to see how ScaleGMNs perform on different tasks, especially an equivariant task (rather than just invariant tasks) such as INR editing.
2. ScaleInv is oddly described. The equation on Page 6 should probably have $\tilde x_i$ or something similar as its arguments, instead of $x_i$ (because if the $\rho^k$ are just general MLPs as you say right after the equation, then this is not scale invariant).
3. At the end of Page 6 and beginning of Page 7, you say that sign canonicalization can only be used for dimension 1, but this is not quite true. In Ma et al. 2023 and Ma et al. 2024, algorithms are given for canonicalizing with respect to the sign group, for use on inputs to a neural network.

References
* [Ma et al. 2023] Laplacian Canonization: A Minimalist Approach to Sign and Basis Invariant Spectral Embedding. https://arxiv.org/abs/2310.18716
* [Ma et al. 2024] A Canonization Perspective on Invariant and Equivariant Learning.
 https://arxiv.org/abs/2405.18378

**Questions:**

1. Do you have a way to handle translation symmetries in nonlinearities?

**Limitations:**

Limitations are discussed on Page 9

---

> ### Author Rebuttal · Authors · 2024-08-06
>
> ### Weaknesses
>
> >*Weakness 1. It would be interesting to see how ScaleGMNs perform on different tasks, especially an equivariant task (rather than just invariant tasks) such as INR editing.*
>
>
> Metanetworks can indeed find interesting applications that require our model to be permutation and scale equivariant. Please refer to the global response for our experiments on INR editting and the corresponding results in Table 1 of the attached PDF. The bidirectional variant of our method, ScaleGMN-B, is able to surpass all the GNN-based metanetworks, even the ones using probe features.
>
> ---
>
> >*Weakness 2. ScaleInv is oddly described. The equation on Page 6 should probably have or something similar as its arguments, instead of (because if the are just general MLPs as you say right after the equation, then this is not scale invariant).*
>
> Thank you for spotting this. Indeed, there is a typo in the definition of ScaleInv (below L265). The arguments of the function $\rho^k$ (universal approximators - MLPs) should have been $\tilde{\mathbf{x}}_i$ instead of $\mathbf{x}_i$, where $\tilde{\mathbf{x}}_i$ are explained later in the text (L277) and are the outputs of a canonicalisation or a symmetrisation function, i.e. $\tilde{\mathbf{x}}_i = \text{canon}(\mathbf{x}_i)$ or $\tilde{\mathbf{x}}_i = \text{symm}(\mathbf{x}_i)$, which ensures invariance. We will fix this in an updated version of the manuscript.
>
> ---
>
> >*Weakness 3. At the end of Page 6 and beginning of Page 7, you say that sign canonicalization can only be used for dimension 1, but this is not quite true. In Ma et al. 2023 and Ma et al. 2024, algorithms are given for canonicalizing with respect to the sign group, for use on inputs to a neural network.*
>
> We are grateful to the reviewer for bringing up these two recent references that deal with sign canonicalisation - we were not aware of these works and, indeed, they are useful for our setup. Given the fact that symmetrisation introduces additional parameters (i.e. the internal MLP, see L276), we are interested in conducting experiments in the future with the proposed canonicalisation method, so as to examine if similar performance can be achieved with a reduced parameter count. Additionally, we will update our text accordingly to complement our discussion with this missing point.
>
>
> ### Questions
>
> >*Question 1. Do you have a way to handle translation symmetries in nonlinearities?*
>
>
> Translation symmetries (such as those induced  by the softmax activation)  are an important next step in this research direction. Our method currently does not handle this case. A potentially straightforward modification might be to follow the same rationale with scale equivariant networks: first, define a translation invariant module via canonicalisation and second, use it to achieve equivariace (e.g.  translate the input by the output of the invariant module). For example, for symmetries of the form $\mathbf{x}’ =  \mathbf{x} + a$, where $a$ is a scalar, we can canonicalise as follows  $\tilde{\mathbf{x}} =  \mathbf{x} - \frac{1}{N} \sum_{i=1}^N x_i $.
>
> We believe that characterizing even more nonlinearities (or families of them) and designing the respective invariant modules, is a prosperous future work towards implementing a unified framework able to handle various types of networks.

---

> > ### Comment · Reviewer_AXwX · 2024-08-08
> >
> > We thank the authors for their rebuttal. The new experimental results are also very strong.
> >
> > I maintain my score of 8, and definitely support acceptance of this paper.

---

### Author Rebuttal · Authors · 2024-08-07

We would like to thank the reviewers for their thorough evaluation of our paper and their constructive feedback, which helped us improve our empirical evaluation to further corroborate our claims and identify potential future directions. In the following comments, we gather the strengths pointed out by the reviewers and summarise our rebuttal response and changes that will be made in an updated version of the manuscript.

- **Rev. AXwX** *found our paper well-written* (AXwX: "Great writing. Nice introduction and related work, as well as good setup and notation in Section 3."),
- **Rev. x2pG, ugM4** *underlined the significance of the studied problem* (x2pG: "[...] this paper tackles a crucial field and problem, advancing the area in a way that can benefit many practitioners.", ugM4: "The paper deals with an important and timely problem of learning in deep weight spaces.").
- **Rev. AXwX, x2pG, ugM4** *acknowledged the novelty of our method* (AXwX: "Their ScaleGMNs extend metanetworks, which are typically only permutation equivariant (if at all equivariant), to also account for other symmetries in input neural networks' parameters.", x2pG: "[...] a new structure that incorporates both permutation and scale symmetries [...].", ugM4: "[...] presents a novel architecture by incorporating scale and permutation equivariance.").
- **Rev. AXwX, x2pG, ugM4** *acknowledged the importance of our theoretical contributions regarding the expressive power of ScaleGMN* (AXwX: "Nice theoretical results. That ScaleGMN can express the forward and backward pass is a good way to check that its expressive power is not overly limited when adding the additional scaling equivariances.", x2pG: "The authors [...] show that ScaleGMN can simulate forward and backward passes of arbitrary inputs in the processed NN.", ugM4: "The paper provides theoretical analysis and results regarding the expressive power of the proposed approach."
- **Rev. AXwX, x2pG, ugM4** *appreciated the empirical impovements reported in our experimental evaluation* (AXwX: "Large empirical improvements, especially on INR classification.", x2pG: "The empirical results show a significant improvement compared to recent works in the field of weight space learning.", ugM4: "Empirical results show significant improvement over baseline methods.").
- In addition to the above, **Rev. AXwX** *pinpoints that these improvements are achieved with built-in equivariance alone and without having to resort to additional practical tricks* (AXwX: "without many of the unfair or expensive tricks that others use! [...] That ScaleGMN can beat the prior methods without these tricks is very impressive. [...] ScaleGMN does not need random Fourier features or data augmentations")

## Rebuttal Summary

### Equivariant tasks
Reviewers AXwX, x2pG, ugM4 correctly mention the need for an equivariant task. To that end, we evaluate on **INR editing**. Following [2], we select the MNIST-INR dataset and evaluate our method. Again, no additional tricks or augmentations were used.

**Results**. (Table 1) Bidirectional *ScaleGMN-B* achieves an MSE test loss ($10^{-2}$) equal to $1.89$, **surpassing all baselines**, outperforming even the *NG-GNN baseline with 64 probe features*. Note that the performance gap between the bidirectional and the forward model (which achieves a loss of $2.56$) is expected for equivariant tasks: in this case we are required to compute representations for every graph node, yet, in the forward variant, the earlier the layer of the node, the less information it receives. Similarly, our baselines are either bidirectional (NG-GNN [2]) or non-local (DWS [3], NFN [1]).

### Heterogeneous activation functions
Reviewer x2pG  pointed to heterogeneous activation functions. *In principle, our method does not impose any limitations regarding their homogeneity*.

**Results**. Evaluated on CIFAR-10-GS, **ScaleGMN demonstrates superior performance** compared to the baselines, *significantly surpassing the next best model*. (Table 4)

### Scaling data augmentations
Reviewer ugM4 proposed baselining with a permutation-equivariant-only model combined with random scaling augmentations.

**Results.** (Table 3)
* **Sign symmetries**: We augment w/  sign flips **independently** (probability 0.5). This surpasses the original baselines, but **not ScaleGMN and ScaleGMN-B.**
* **Positive-scale symmetries**: We sample positive scalars  (NB: **continuous and unbounded distribution**), but observe performance deterioration compared to the original baselines.

### More complex architectures
We acknowledge that experimenting with diverse architectures would strengthen our contributions. However, there are two reasons why this was not possible in the current work. First, the characterisation of scaling symmetries holds for MLPs and can be extended to CNNs. However, extending it to other architectures requires further efforts and is therefore more appropriate to be considered in future work.

Second, weight-space learning is currently missing curated benchmarks of complex architectures. Lim et al. [4] and Zhou et al. [5] experiment with diverse architectures, using private datasets. Small DNN Zoo [6] and ModelZoos [7] contain trained CNNs of fixed architectures. Finally, the diverse CNN Wild Park [2] was only made public a few days ago. *Keeping in mind that processing such networks is not among our main contributions*, we opted to align with the previous works on metanetworks and selected the datasets used in [1], [2].

### Exposition/writing style
We thank the reviewers x2pG and ugM4 for their suggestions. We allocated a good portion of the paper to problem formulation and background, as the topic of weight space learning is relatively new. The characterisation of scaling symmetries is also recent and has not yet gathered significant attention. Consequently, we opted for a smooth and detailed introduction before delving into our contributions to make our work self-sufficient.

---

### Author Response · Authors · 2024-08-07
**References for Author Rebuttal**

[1] Zhou, Allan, et al. "Permutation equivariant neural functionals." Advances in neural information processing systems 36 (2024).

[2] Kofinas, Miltiadis , et al. "Graph neural networks for learning equivariant representations of neural networks." ICLR, 2024.

[3] Navon, Aviv, et al. "Equivariant architectures for learning in deep weight spaces." International Conference on Machine Learning. PMLR, 2023.

[4] Lim, Derek, et al. "Graph Metanetworks for Processing Diverse Neural Architectures." The Twelfth International Conference on Learning Representations.

[5] Zhou, Allan, Chelsea Finn, and James Harrison. "Universal neural functionals." arXiv preprint arXiv:2402.05232 (2024).

[6] Unterthiner, Thomas, et al. "Predicting neural network accuracy from weights." arXiv preprint arXiv:2002.11448 (2020).

[7] Schürholt, Konstantin, et al. "Model zoos: A dataset of diverse populations of neural network models." Advances in Neural Information Processing Systems 35 (2022): 38134-38148.

---

### Decision · Program_Chairs · 2024-09-25

**Decision:**

Accept (oral)

**Comment:**

This paper introduces Scale Equivariant Graph Meta Networks (ScaleGMNs), extending metanetwork design to include scaling symmetries. The work presents strong theoretical analysis and empirical improvements in the emerging field of deep weight space learning.
Initial concerns about experimental scope and presentation were adequately addressed in the authors' rebuttal. They provided additional experiments and committed to improving the paper's structure.
Given the novel contributions, theoretical foundation, and empirical results, I recommend accepting this paper for NeurIPS 2024. The authors should incorporate the promised improvements in the camera-ready version.